# Asynchronous Decentralized SGD with Quantized and Local Updates

**Giorgi Nadiradze**
IST Austria
giorgi.nadiradze@ist.ac.at

**Amirmojtaba Sabour**
IST Austria
amsabour79@gmail.com

**Peter Davies**
University of Surrey
pd0034@surrey.ac.uk

**Shigang Li**
ETH Zurich
shigangli.cs@gmail.com

**Dan Alistarh**
IST Austria & Neural Magic
dan.alistarh@ist.ac.at

## Abstract

Decentralized optimization is emerging as a viable alternative for scalable distributed machine learning, but also introduces new challenges in terms of synchronization costs. To this end, several communication-reduction techniques, such as non-blocking communication, quantization, and local steps, have been explored in the decentralized setting. Due to the complexity of analyzing optimization in such a relaxed setting, this line of work often assumes *global* communication rounds, which require additional synchronization. In this paper, we consider decentralized optimization in the simpler, but harder to analyze, *asynchronous gossip* model, in which communication occurs in discrete, randomly chosen pairings among nodes. Perhaps surprisingly, we show that a variant of SGD called *SwarmSGD* still converges in this setting, even if *non-blocking communication*, *quantization*, and *local steps* are all applied *in conjunction*, and even if the node data distributions and underlying graph topology are both *heterogenous*. Our analysis is based on a new connection with multi-dimensional load-balancing processes. We implement this algorithm and deploy it in a super-computing environment, showing that it can outperform previous decentralized methods in terms of end-to-end training time, and that it can even rival carefully-tuned large-batch SGD for certain tasks.

## 1 Introduction

Decentralized optimization has recently emerged as a promising approach for scaling the distributed training of machine learning models, in particular via stochastic gradient descent (SGD) [Lian et al., 2017, Tang et al., 2018, Koloskova et al., 2019a]. Its key advantage is that it removes the need for a central coordinator node in distributed training, and therefore it can allow for high scaling.

The general decentralized optimization setting is the following: we are given $n$ nodes, each with a subset of data from some distribution, which can communicate over some underlying graph topology. In each global round, each node samples some local data, performs a local gradient step, and it is paired with a neighbor, which may be chosen randomly. The nodes exchange model information pairwise, and then update their models, often via direct model averaging. Variants of this setting have been analyzed since pioneering work by Tsitsiklis [1984], for various estimation and optimization algorithms [Xiao and Boyd, 2004, Nedic and Ozdaglar, 2009, Johansson et al., 2009, Shamir and Srebro, 2014] and have seen renewed interest given its applicability to training deep neural networks (DNNs) at scale, e.g. [Lian et al., 2017, 2018, Assran et al., 2018].

Recently, there has been significant focus on reducing the *synchronization overheads* for decentralized training, usually employing three approaches: 1) implementing faster *non-blocking communication* between communication partners at a round [Lian et al., 2018, Assran et al., 2018], which may cause them to see stale versions of their models, 2) allowing nodes to take *local steps* in between

their communication rounds [Wang and Joshi, 2018, Koloskova et al., 2020], and 3) applying *quantization* to the communication [Lu and De Sa, 2020, Tang et al., 2018, Koloskova et al., 2019a,b].

The above impressive line of work contributes a rich set of algorithmic and analytic ideas; however, one common limitation is that the algorithms are usually set in the *synchronous gossip* model, which requires all nodes to perform their communication in lock-step rounds, and share a common notion of time, thus reducing their practicality. To mitigate this fact, some references, e.g. [Lian et al., 2018, Assran et al., 2018, Lu and De Sa, 2020] partially relax this requirement, although they do so at the cost of additional assumptions, or reduced guarantees, as we discuss in related work. Another relative limitation is that the analyses are usually customized to the bespoke communication-reduced methods being applied, and therefore are hard to generalize to other methods.

**Our Contribution.** In this paper, we consider decentralized SGD-based optimization in the simpler, but harder to analyze, *asynchronous gossip* model [Xiao and Boyd, 2004], in which communication occurs in discrete, randomly chosen pairings among nodes, and does not require a common notion of time. We prove that a new variant of SGD we call *SwarmSGD* converges in this setting, even though it supports all three communication-reduction approaches mentioned above *in conjunction*. Our analysis generalizes to heterogeneous data distributions and communication topologies.

At a high level, SwarmSGD works as follows. Each node $i$ maintains a local model estimate $X_i$ based on which gradients are generated, and a shared buffer where quantized models are stored for communication with other nodes. In each step, node $i$ first computes a sequence of $H$ local gradient steps, which it does not yet apply. Next, the node chooses communication partner $j$, uniformly at random among its neighbors. Then, node $i$ reads from its own communication buffer and from the communication buffer of $j$, obtaining quantized models $Q_i$ and $Q_j$. A subtlety here is that $Q_i$ is not necessarily the quantized version of the model $X_i$, since other nodes can write concurrently to $i$'s buffer. The node $i$ then averages $Q_i$ with $Q_j$, and updates the neighbor's remote buffer to the quantized average. Finally, it applies its local gradient steps to the resulting average, adopts this as its next model $X_i$, and a writes quantized version of it in its own shared buffer. This procedure can be implemented in a deadlock-free, non-blocking manner, by using either shared-memory or the remote direct-memory access (RDMA) calls supported by MPI [Woodall et al., 2006]. Importantly, the communication partner $j$ does not need to block its computation during communication, and may be contacted by more than one interaction partner during a single local step, although we do assume that individual reads and writes are performed atomically.

A key component of this procedure is the *quantization scheme*: directly using an unbiased quantizer, e.g. [Alistarh et al., 2017] would destroy convergence guarantees, as the quantization error would be proportional to the model norm, which may not be bounded. Instead, we use a customized variant of the quantization scheme of Davies et al. [2021], whose error depends on the distance between the *point being quantized* (the model), and an arbitrary *reference point*, provided as a parameter. We prove that each node can reliably use *its own model* as a *reference point* to quantize and de-quantize messages placed in its buffer by other nodes. In turn, this requires care in the analysis.

Specifically, the key observation behind our analysis is exactly in showing that the nodes' local models stay well-enough concentrated around their mean throughout optimization to allow for correct decoding of quantized models, which in turn implies joint convergence by the nodes towards a point of vanishing gradient. This concentration follows via a non-trivial super-martingale argument. If nodes take a constant number of local SGD steps between communication steps, then SwarmSGD has $\Theta(\sqrt{n})$ speedup to convergence for non2-convex objectives. This matches results from previous work which considered decentralized dynamics but with global synchronization [Lian et al., 2017].

**Experimental Validation.** We apply SwarmSGD to train deep neural networks on image classification and machine translation (NMT) tasks, deployed on the Piz Daint supercomputer [Piz, 2019]. Experiments confirm the intuition that the average synchronization cost of SwarmSGD per iteration is low: it stays at less than $10\%$ of the batch computation time, and remains constant as we increase the number of nodes. For example, using SwarmSGD, we are able to train a Transformer-XL [Vaswani et al., 2017] model on WMT17 (En-Ge) $1.5\times$ faster than a highly-optimized large-batch SGD baseline, and to *slightly higher accuracy*, without additional hyper-parameter tuning. At the same time, due to the reduced communication frequency, Swarm also improves upon the speed of the previous practical decentralized methods, e.g. [Lian et al., 2017, 2018, Assran et al., 2018]. Importantly, we also note that, in less overparametrized settings such as training residual CNNs [He et al., 2016] on ImageNet [Russakovsky et al., 2015], nodes do need to perform more iterations over the dataset relative to the baseline in order to recover full accuracy. This is predicted by the analysis, and confirms similar findings in previous work [Assran et al., 2018]. Overall, our method does

appear well-suited to training large modern models at node counts where global synchronization among all nodes is prohibitively expensive.

**Related Work.** Decentralized optimization has a long history [Tsitsiklis, 1984], and is related to the study of gossip algorithms, e.g. [Kempe et al., 2003, Xiao and Boyd, 2004, Boyd et al., 2006]. Gossip is usually studied in one of two models [Boyd et al., 2006]: *synchronous*, structured in global rounds, where each node interacts with a randomly chosen neighbor, forming a matching, and *asynchronous*, where each node wakes up at random times, e.g. given by a Poisson clock, and picks a random neighbor to interact with. Several classic optimization algorithms have been analyzed in the *asynchronous gossip model* [Nedic and Ozdaglar, 2009, Johansson et al., 2009, Shamir and Srebro, 2014]. In this paper, we focus on analyzing decentralized SGD in this model.

As mentioned, the growing line of work on decentralized optimization for machine learning has mostly focused on variants of the *synchronous* gossip model. Specifically, Lian et al. [2017] considered this setting in the context of DNN training, while and Tang et al. [2018] and Koloskova et al. [2019b] also analyzed decentralized optimization with quantization in the *synchronous* model. Wang and Joshi [2018] and Koloskova et al. [2020] provided analysis frameworks for synchronous decentralized SGD with local updates, and possibly changing topologies.

Lian et al. [2018] and Assran et al. [2018] focused specifically on reducing *synchronization* costs in this setting, and proposed algorithms with *partially non-blocking communication*, in which nodes may read a *stale* version of the interaction partner's information, modelling e.g. a communication buffer. However, the maximum staleness must be bounded by a global variable $\tau$, which must be enforced throughout the execution. As observed by Assran et al. [2018], enforcing this bound can cause *blocking*, and therefore the authors of these works propose to implement a relaxed round-based model, in which nodes interact once per round in perfect matchings. Their algorithms provide $O(1/\sqrt{Tn})$ convergence rates, under analytical assumptions.

Upon careful examination, we find that their analysis approach can be extended to the asynchronous gossip model we consider, by defining the "contact matrices" to correspond to pairwise interactions. However, this introduces two significant limitations. First, the analysis will not support *local gradient updates to models* nor *quantized communication*. If we remove these practical relaxations, our technique yields better bounds, as our potential analysis is specifically-tailored to this dynamic interaction model. Second, as we detail in the Appendix, some of their technical conditions imply existence of global synchronization. For Assran et al. [2018], as we detail in the Appendix, their analysis would not guarantee any non-trivial speedup due to parallelization in the asynchronous gossip model. We describe these issues in detail and present a systematic comparison in Appendix A.

Lu and De Sa [2020] provided a novel approach to analyze decentralized SGD with quantization and *limited asynchrony*: specifically, their algorithm requires *blocking* communication, i.e. nodes have to synchronize explicitly during interactions, but may see old versions of eachothers' models. More precisely, during each interaction, both parties are responsible for updating their local models, meaning that once node is woken up (we call it initiator node) and chooses interaction partner it has to block until the partner is woken up as well. In our case, initiator can update both its local model and the local model of its partner and proceed to the next step without blocking. Koloskova et al. [2019a] use a similar update rule in the synchronous model. Zhang and You [2021] recently proposed a decentralized algorithm which is fully-asynchronous as long as node activation rates and message delays are bounded. As noted earlier, bounding activation rates does imply blocking; however, tolerating (bounded) message delays does improve over our approach of updating models using atomic writes. The setting further differs in that they assume that nodes compute full (non-stochastic) gradients, as well as that the loss function satisfies the PL condition.

In sum, we are the first to explicitly consider the *asynchronous gossip model*, and the impact of local updates, asynchrony, and quantization used *in conjunction* together with decentralized SGD. Our technique is new, relies on a fine-grained analysis of individual interactions, and can yield improved bounds even in the case where $H = 1$. Further, our algorithm is the first to allow for both communication-compression and non-blocking communication. From the implementation perspective, the performance of our algorithm matches or improves that of previous methods, notably D-PSGD [Lian et al., 2017], AD-PSGD [Lian et al., 2018] and SGP [Assran et al., 2018].

## 2 Preliminaries

**The Distributed System Model.** We consider a model which consists of $n \geq 2$ nodes, each of which is able to perform local computation. We assume that communication network of nodes is

a graph $G$ with spectral gap $\lambda_2$, which denotes the second smallest eigenvalue of the Laplacian of $G$. Let $\rho_{max}, \rho_{min}$ be the maximum and minimum degrees in $G$, respectively. We will focus on densely-connected topologies, which model supercomputing and cloud networks: for instance, the standard Dragonfly topology [Kim et al., 2008, Besta and Hoefler, 2014] is regular, densely connected and low-diameter, mimicking regular expanders.

The execution is modelled as occurring in discrete *steps*, where in each step a new node (the "initiator") is sampled, and can then contact one of its neighbors (the "responder") uniformly at random. (At the algorithm level, the initiator is "sampled" once it completes its current computational step, and seeks to interact with a neighbor.) We denote the number of steps for which we run by $T$. Globally, the communication steps can be seen as a sequence of sampled *directed* communication edges. Thus, the basic unit of time is a single pairwise interaction between two nodes. Notice however that in a real system $\Theta(n)$ of these interactions could occur in parallel. Thus, the standard global time measure is *parallel time*, defined as the total number of interactions divided by $n$, the number of nodes. Parallel time intuitively corresponds to the *average* number of interactions per node until convergence. This model is identical to the asynchronous gossip model [Xiao and Boyd, 2004], and to the population protocol model [Angluin et al., 2006].

**Stochastic Optimization.** We assume that the agents wish to jointly minimize a $d$-dimensional, differentiable function $f : \mathbb{R}^d \to \mathbb{R}$. Specifically, we will assume the empirical risk minimization setting, in which agents are given access to a set of $m$ data samples $S = \{s_1, \ldots, s_m\}$ coming from some underlying distribution $\mathcal{D}$, and to functions $\ell_i : \mathbb{R}^d \to \mathbb{R}$ which encode the loss of the argument at the sample $s_i$. The goal of the agents is to converge on a model $x^*$ which minimizes the empirical loss over the $m$ samples, that is $x^* = \operatorname{argmin}_x f(x) = \operatorname{argmin}_x (1/m) \sum_{i=1}^{m} \ell_i(x)$. We assume that each agent $i$ has a local function $f_i$ associated to its fraction of the data, i.e $\forall x \in \mathbb{R}^d$: $f(x) = \sum_{i=1}^{n} f_i(x)/n$.

Agents employ these samples to run a decentralized variant of SGD, described in detail in the next section. For this, we will assume that each agent $i$ has access to *unbiased stochastic gradients* $\widetilde{g}_i$ of the function $f_i$, which are functions such that $\mathbb{E}[\widetilde{g}_i(x)] = \nabla f_i(x)$. Stochastic gradients can be computed by each agent by sampling i.i.d. the distribution $\mathcal{D}$, and computing the gradient of $f$ at $\theta$ with respect to that sample. Our analysis also extends to the case where each agent is sampling from its own partition of data. We assume the following conditions about the objective function, although not all our results require the second moment bound:

1. **Smooth Gradients**: The gradient $\nabla f_i(x)$ is $L$-Lipschitz continuous for some $L > 0$, i.e. for all $x, y \in \mathbb{R}^d$ and agent $i$:
$$\|\nabla f_i(x) - \nabla f_i(y)\| \leq L\|x - y\|. \tag{1}$$

2. **Bounded Variance**: The variance of the stochastic gradients is bounded by some $\sigma^2 > 0$, i.e. for all $x \in \mathbb{R}^d$ and agent $i$:
$$\mathbb{E}\left\|\widetilde{g}_i(x) - \nabla f_i(x)\right\|^2 \leq \sigma^2. \tag{2}$$

3. **Bounded Local Function Variance**: There exists $\varsigma^2 > 0$, such that for all $x \in \mathbb{R}^d$:
$$\sum_{i=1}^{n} \frac{\left\|\nabla f(x) - \nabla f_i(x)\right\|^2}{n} \leq \varsigma^2. \tag{3}$$

4. **Bounded Second Moment**: The second moment of the stochastic gradients is bounded by some $M^2 > 0$, i.e. for all $x \in \mathbb{R}^d$ and agent $i$:
$$\mathbb{E}\left\|\widetilde{g}_i(x)\right\|^2 \leq M^2. \tag{4}$$

Note that throughout this paper for any random variable $X$, by $\mathbb{E}\|X\|^2$ we mean $\mathbb{E}[\|X\|^2]$.

Each node has a communication buffer, which, for simplicity, we assume can be read and written atomically by each node; Importantly, buffers can only hold *quantized* quantized vectors.

**Quantization Procedure.** We use a quantization function which follows from Lemma 23 in (the full version of) Davies et al. [2021].

**Corollary 2.1.** *(Quantization for Communication Buffers) Fix parameters $R$ and $\epsilon > 0$. There exists a quantization procedure defined by an encoding function $Enc_{R,\epsilon} : \mathbb{R}^d \to \{0, 1\}^*$ and a decoding function $Dec_{R,\epsilon} = \mathbb{R}^d \times \{0, 1\}^* \to \mathbb{R}^d$ such that, for any vector $x \in \mathbb{R}^d$ which we are trying to quantize, and any vector $y$ which is used by decoding, which we call the* decoding

key, *if* $\|x - y\| \leq R^{R^d}\epsilon$ *then with probability at least* $1 - \log\log(\frac{\|x-y\|}{\epsilon})O(R^{-d})$, *the function* $Q_{R,\epsilon}(x) = Dec_{R,\epsilon}(y, Enc_{R,\epsilon}(x))$ *has the following properties:*

1. *(Unbiased decoding)* $\mathbb{E}[Q_{R,\epsilon}(x)] = \mathbb{E}[Dec_{R,\epsilon}(y, Enc_{R,\epsilon}(x))] = x$;

2. *(Error bound)* $\|Q_{R,\epsilon}(x) - x\| \leq (R^2 + 7)\epsilon$;

3. *(Communication bound)* *To compute* $Dec_{R,\epsilon}(y, Enc_{R,\epsilon}(x))$, *only the first $B$ bits of* $Enc_{R,\epsilon}(x)$ *are needed, where* $B = O\left(d\log(\frac{R}{\epsilon}\|x - y\|)\right)$.

*Proof.* Lemma 23 of the full version of Davies et al. [2021] provides similar guarantees as the ones we want to prove, but they assume interactive message-passing communication between an encoding node $u$ and a decoding node $v$. However, in their setting, the messages sent by $u$ are *non-adaptive*: $u$ simply sends quantizations using an increasing number of bits, until $v$ replies confirming that it has decoded successfully. The number of bits sent during communication is upper bounded by $O\left(d\log(\frac{R}{\epsilon}\|x - y\|)\right)$, where $x$ is a vector node $u$ is sending and $y$ is vector node $v$ is using for decoding. In our setting, we use communication buffers which, so node $u$ can simply append all of its potential messages together as $Q_{R,\epsilon}(x)$.

Critically, notice that node $u$ should append enough bits so that the decoding is possible (Since in our setting there is no way for $v$ to acknowledge that it received enough number of bits). This can be done in two ways. If $u$ knows the distance between $x$ and $y$. then $u$ can simply write $O\left(d\log(\frac{R}{\epsilon}\|x - y\|)\right)$ bits in the register.

In the second case, $u$ does not know the distance. Let $T$ be the total number of times nodes communicate throughout our algorithm. We will show that with high probability all distances between encoded and decoding vectors will be at most $\frac{\epsilon T^{17}}{R}$ (dependence on $T$ stems from the fact that we wish to show an upper bound with high probability, please see Lemma B.19 in the Appendix), and therefore at most $O(d\log T)$ bits for quantization will suffice in the worst case. Thus, the node writes $O(d\log T)$ bits in the register, but when $v$ tries to decode, it does not need all those bits: it reads and uses only the first $O\left(\log(\frac{R}{\epsilon}\|x - y\|)\right)$ bits.

**Counting Communication Cost.** We emphasize that, when we calculate the number of bits needed by quantization we actually aim to measure the number of bits *exchanged between* $u$ and $v$. In the setting we consider, which has local registers/communication buffers, this is the number of bits spent to read from (or to write to) the non-local register. Since the second case above involves writing a relatively large number of bits, we will use it only when $u$ is writing a quantized value to its *own* register/buffer, and so does not need to communicate the bits. Then, only the $O\left(\log(\frac{R}{\epsilon}\|x - y\|)\right)$ bits read by $v$ need to be communicated.

To summarize, in our algorithm we will always ensure that whenever some node $u$ writes a quantized value, it either knows the key which will be used for decoding it, or is writing to its local register. $\square$

## 3 The SwarmSGD Algorithm

We now describe a decentralized variant of SGD, designed to be executed by a population of $n$ nodes, interacting over the edges of communication graph $G$, as described above. The algorithm proceeds in individual communication steps, where in each step a node which has completed its local computation, seeks a random neighbor to communicate with. We will alternatively say that node gets activated (once it finished computation) and then becomes initiator of the interaction.

**The Communication Registers.** The communication buffer of each node $i$ consists of two registers: one containing an encoded version of its own (possibly outdated) model, which will only be written to by node $i$ itself, and one for holding an encoded version of its current model, which will only be written to by other nodes. (This second register can be seen as a "communication queue" for the nodes wishing to communicate with $i$.) Initially all registers contain zero vectors.

**Parallel Execution.** For simplicity we will skip the details of quantization, and assume that nodes write and read quantized models directly, without encoding and decoding steps. Both current and outdated models are zero vectors initially. Each node $i$ computes a number of local gradients based on its last model view, and other nodes may update this model while $i$ is computing. Hence, only after node $i$ is done with computing local gradients does it read its updated current model. Let $\hat{X}_i$ be the value of the outdated model and let $X_i$ be the value of the current model. Node $i$ computes the average of quantized models $\frac{Q(X_i) + Q(X_j)}{2}$ and writes it in a register which contains current model

of node $j$. Next, it computes $\frac{Q(X_i)+Q(X_j)}{2} - \eta\widetilde{h}(\hat{X}_i)$ (where $\eta$ is a learning rate and $\widetilde{h}(\hat{X}_i)$ is a sum of local gradients), and writes it in both of its local registers, one containing the current model and one containing the outdated model. Once the write is finished, it again proceeds to compute local gradients, based on the view $\frac{Q(X_i)+Q(X_j)}{2} - \eta\widetilde{h}(\hat{X}_t^i)$.

**Sequential model.** For the analysis, it is useful to map these parallel interactions to a sorted sequence of sequential ones. Thus, time steps track the interactions between agents, and each interaction consists of random number of local steps steps which activated node performs, plus one averaging step where activated node (or initiator node) contacts its random neighbour. The analysis assumes that nodes get activated randomly, by independent Poisson clocks, which leads to a uniform global sampling distribution. In practice, this could be approximated by having the number of local gradient steps executed by each node be a geometric random variable of mean $H$. For the sake of practicality, our experiments will take $H$ to be a small constant, instead of a random variable, which yields similar results. The pseudocode from the point of view of a single node $i$ which was activated at step $t+1$ is given in Algorithm 1.

For $t \geq 0$, let $Enc(\hat{X}_t^i)$ and $Enc(X_t^i)$ be the values written in the registers containing the outdated and the current model of agent $i$ after $t$ steps, respectively. That is, $X_t^i$ is the current model of agent $i$ and $\hat{X}_t^i$ is the outdated model.

**The Communication Procedure.** Since $i$ was activated at step $t$ we will assume that it has already computed $H_i$ local gradients using the outdated model $\hat{X}_t^i$, where $H_i$ is a geometric random variable with mean $H$, as follows. Let $\widetilde{h}_i^0(\hat{X}_t^i) = 0^d$; for indices $1 \leq q \leq H_i$, let $\widetilde{h}_i^q(\hat{X}_t^i) = \widetilde{g}_i(\hat{X}_t^i - \sum_{s=0}^{q-1} \eta\widetilde{h}_i^s(\hat{X}_t^i))$ be the $q$-th local gradient. Then, let $\widetilde{h}_i(\hat{X}_t^i) = \sum_{q=1}^{H_i} \widetilde{h}_i^q(\hat{X}_t^i)$ be the sum of all computed local gradients. Or alternatively, since we are in a sequential setting, we can assume that $i$ does computation at step $t$. First, $i$ retrieves $Q(X_t^i)$ (the quantized version of its current model), by decoding $Enc(X_t^i)$ using key $Q(\hat{X}_t^i)$. We would like to note that $i$ can obtain $Q(\hat{X}_t^i)$ simply by decoding $Enc(\hat{X}_t^i)$, using key $\hat{X}_t^i$ (which it knows, to full precision, since it calculated the value itself), and this step does not cost any communication bits since all of the terms involved are local to $i$'s registers.

Then, it contacts its interaction partner $j$. Node $i$ calculates $Q(\hat{X}_t^j)$ by decoding $Enc(\hat{X}_t^j)$, again using $\hat{X}_t^i$ as a key, and then it retrieves $Q(X_t^j)$ by decoding $Enc(X_t^j)$ with key $Q(\hat{X}_t^j)$. Then, $i$ calculates $X_{t+1}^i = \frac{Q(X_t^i)}{2} + \frac{Q(X_t^j)}{2} - \eta\widetilde{h}_i(\hat{X}_t^i)$ and $X_{t+1}^j = \frac{Q(X_t^i)}{2} + \frac{Q(X_t^j)}{2}$.

Next, node $i$ calculates $Enc(X_{t+1}^i)$ and writes to its own register for its outdated models. Here, we use the first case for quantization using Corollary 2.1: $i$ is not aware of the key that other nodes will use for decoding, but since it is writing to its own local register, it can afford to use the worst-case $O(d \log T)$ bits. Additionally, it writes $Enc(X_{t+1}^i)$ to its own register containing current model, so that there are enough bits for $Q(\hat{X}_{t+1}^i)$. (Note that $\hat{X}_{t+1}^i = X_{t+1}^i$ has to be used as decoding key.)

Finally, it calculates $Enc(X_{t+1}^j)$ and writes it in the register which contains the current model of $j$, using enough bits that it can be decoded using $Q(\hat{X}_{t+1}^j)$ (we have that $\hat{X}_{t+1}^j = \hat{X}_t^j$). Notice that, the way our algorithm is specified, every node which tries to decode $Enc(X_{t+1}^j)$ will use $Q(\hat{X}_{t+1}^j)$ as a key (which $i$ knows), hence Corollary 2.1 holds in this case as well. We emphasize the fact that all this communication is one-way, as it does not require $j$'s intervention. By Corollary 2.1 the total number of bits used is :

$$O\Big(d\log(\frac{R}{\epsilon}\|\hat{X}_t^i - \hat{X}_t^j\|)\Big) + O\Big(d\log(\frac{R}{\epsilon}\|Q(\hat{X}_t^j) - X_t^j\|)\Big) + O\Big(d\log(\frac{R}{\epsilon}\|Q(\hat{X}_t^j) - X_{t+1}^j\|)\Big).$$

(Recall that we count only reading and writing to *other* registers, and do not count operations $i$ performs on its *own* registers.)

We will show that we can make the probability of any instance of quantization failing less than $1/T^c$, for some sufficiently large constant $c$, by setting the constant factor in the number of bits sufficiently high. Then, we can take a union bound over all instances of quantization throughout the algorithm, to show that none fail with high probability in $T$. Henceforth, we will then be able to prove the convergence of our algorithm conditioned on this event.

**Avoiding race conditions.** An interesting question is what happens when *multiple nodes* contact $j$ concurrently. For conciseness, our pseudocode assumes that the update sequence in lines 8–

---
**Algorithm 1** Sequential SwarmSGD pseudocode for each interaction between nodes $i$ and $j$.
---
1: % Let $G$ be a communication graph.
2: % Initial models $X_0^1 = X_0^2 = ... = X_0^n$
3: **for** $t = 0$ **to** $T - 1$ **do**
4:     Sample the initiator node $i$ uniformly at random.
5:     Node $i$ samples a node $j$, adjacent to it in $G$, uniformly at random.
6:     Let $t - \tau_t^i$ be the last step at which node $i$ was chosen as initiator.
7:     Let $\hat{X}_t^i = X_{t-\tau_t^i}^i$ be its model from that step.
8:     $Q(X_t^i) \leftarrow Dec(Q(\hat{X}_t^i), Enc(X_t^i))$
9:     $Q(\hat{X}_t^j) \leftarrow Dec(\hat{X}_t^i, Enc(\hat{X}_t^j))$
10:    $Q(X_t^j) \leftarrow Dec(Q(\hat{X}_t^j), Enc(X_t^j))$
11:    $X_{t+1}^i \leftarrow Q(X_t^i)/2 + Q(X_t^j)/2 - \eta\widetilde{h}_i(\hat{X}_{t-1}^i)$
12:    $X_{t+1}^j \leftarrow Q(X_t^i)/2 + Q(X_t^j)/2$
13:    Write $Enc(X_{t+1}^i)$ to the registers containing current and outdated models of node $i$
14:    Write $Enc(X_{t+1}^j)$ to the register containing current model of node $j$
15:    For $k \neq i, j$, $X_{t+1}^k = X_t^k$.
16: **end for**
---

14 happens atomically, but this sequence can cause a data race. To mitigate this, we can use a bounded non-blocking queue [Michael and Scott, 1996] at each node instead of a single buffer. Thus, instead of updating the buffer value atomically, each node simply appends the corresponding quantized model mean to $j$'s communication queue. In practice, this queue is extremely unlikely to be contended, since communication collisions are rare.

## 4   The Convergence of SwarmSGD

Let $\mu_t = \sum_{i=1}^n X_t^i/n$ be the mean over node models at time $t$. Our main result is the following:

**Theorem 4.1.** *Assume the total number of steps $T \geq 10n$, learning rate $\eta = n/\sqrt{T}$, quantization parameters $R = 2 + T^{\frac{3}{d}}$ and $\epsilon = \frac{\eta HM}{(R^2+7)}$. Then, with probability at least $1 - O(\frac{1}{T})$ we have that Algorithm 1 converges at rate*

$$\frac{1}{T}\sum_{t=0}^{T-1}\mathbb{E}\|\nabla f(\mu_t)\|^2 \leq \frac{2(f(\mu_0) - f(x^*))}{H\sqrt{T}} + \frac{6(\sigma^2 + 6H\varsigma^2)}{\sqrt{T}} + \frac{12HM^2}{\sqrt{T}} + C\frac{n^2\rho_{max}^3 H^2 L^2 M^2}{T\rho_{min}\lambda_2^2},$$

*for constant $C$, and uses $O\left(d\log\left(\frac{\rho_{max}^2}{\rho_{min}\lambda_2}\right) + \log T\right)$ expected communication bits per step.*

**Discussion.** First, this notion of convergence is standard in the non-convex case [Lian et al., 2015, 2017, 2018], and each of the upper bound terms has an intuitive interpretation: *the first* represents the reduction in loss relative to the initialization, and gets divided by the number of local steps $H$, since progress is made in this term in every local step; *the second* represents the noise due to stochasticity, and is naturally linear in $H$, as $H$ steps are taken in expectation between two interactions. (Recall that in our model $T$ is the number of interactions, and $TH$ is the expected number of gradient steps.) The *fourth* term encodes overheads caused by local steps, quantization, and graph structure; however, it is usually seen as *negligible* (cf. [Lu and De Sa, 2020]), due to division by $T$.

The third term is the critical one, as it implies a dependence on the second-moment bound. Intuitively, this term appears because our algorithm combines both non-blocking communication, *and* quantization: first, unlike prior work, we do not assume an explicit delay upper bound $\tau$ on communication; in conjunction with quantization, the unbounded delay this implies that our estimate on the model average $\mu_t$ may become dependent on $M$ for large delays, which causes this dependency. While this limitation appears inherent, we are able to remove it if we eliminate quantization: in this case, we get a negligible dependency on $M$. We formalize this in Corollary 4.2.

Second, if we focus on the total number of steps to reach some error bound, we notice an interesting trade-off between the linear reduction in $H$ in the first term, due to local steps, and the linear increase in $H$ in the other terms. Notice that, for dense and low-diameter graphs, such as the regular expanders popular in cluster networks, our convergence bound has no dependence in the graph parameters, and communication is linear in $d$. However, one limitation is that we could have a $\log n$ dependency in the communication for highly-irregular and poorly-connected graphs.

Finally, note that time $T$ here counts *total interactions*. However, $\Theta(n)$ pairwise interactions occur independently in parallel, and so we can slightly abuse notation and replace $T$ by $nT$ in the above formula, to obtain optimal $\Theta(\sqrt{n})$ speedup in terms of wall-clock time. Yet, this speedup is dampened by the variance due to noisy local gradient steps, a fact which we will revisit in the experimental section.

**Proof Overview.** At a high level, the argument rests on two technical ideas. The first is that, in spite of noise and local steps, the nodes' parameters remain concentrated around the mean $\mu_t$. The second is to leverage this, and bound the impact of stochastic noise and model staleness on convergence. In particular, the main technical difficulty in the proof is to correctly "encode" the fact that parameters are well concentrated around the mean. A natural approach is to bound the model variance $\Gamma_t$ after $t$ interactions. Formally, we define $\Gamma_t = \sum_{i=1}^{n} \|X_t^i - \mu_t\|^2$, where $\mu_t = \sum_{i=1}^{n} X_t^i / n$, as before.

We bound the expected evolution of $\Gamma_t$ over time, depending on the learning rate, number of local steps, quantization parameter and the bound provided by the assumption on the stochastic gradients (the bound $M^2$). The critical point is that the upper bound on the expectation of $\Gamma_t$ *does not depend* on the number of interactions $t$. More precisely, if all the above hyper-parameters are constant, we get that $\mathbb{E}[\Gamma(t)] = O(n)$. Our approach brings over tools from classic load-balancing [Berenbrink et al., 2009], to the multi-dimensional case.

Three key elements of novelty in our case are that (1) for us the load balancing process is *dynamic*, in the sense that new loads, i.e. gradients, get continually added; (2) the load-balancing process we consider is multi-dimensional, whereas usually the literature considers simple scalar weights; (3) the models can be *outdated* and *quantized*, which leads to a complex, noisy load-balancing process. We resolve the this third and most challenging issue by using carefully-defined auxiliary potentials.

**Removing the Second-Moment Bound.** Upon reflection, we notice that can render the dependency on $M^2$ negligible if we do not use quantization, but otherwise keep the algorithm the same:

**Corollary 4.2.** *Given the previous assumptions and learning rate $\eta = n/\sqrt{T}$, for some constant $C$, we have that the Algorithm 1 where quantization is the identity converges at rate*

$$\frac{1}{T} \sum_{t=0}^{T-1} \mathbb{E}\|\nabla f(\mu_t)\|^2 \leq \frac{2(f(\mu_0) - f(x^*))}{H\sqrt{T}} + \frac{6(\sigma^2 + 6H\varsigma^2)}{\sqrt{T}} + \frac{Cn^2\rho_{max}^3 H^2 L^2 M^2}{T\rho_{min}\lambda_2^2}.$$

Notice that in this case all the term containing the second moment bound $M^2$ is dampened by a factor of $\frac{1}{T}$, hence we can assume that Algorithm 1 converges at close-to optimal rate $O\left(\frac{2(f(\mu_0) - f(x^*))}{H\sqrt{T}} + \frac{6H(\sigma^2 + 6\varsigma^2)}{\sqrt{T}}\right)$. This result still improves upon previous analyses [Lian et al., 2018, Assran et al., 2018, Lu and De Sa, 2020] in the sense that communication is completely non-blocking (there is no $\tau$), and we allow for local steps.

Further, in the absence of quantization and assuming that the nodes perform single local gradient step, we can entirely remove assumption (4), when $T$ is large enough (e.g for the fully connected graph we will need $T \geq \Omega(n^3)$). More precisely, we can attain convergence rate of $O\left(\frac{f(\mu_0) - f(x^*)}{\sqrt{T}} + \frac{(\sigma^2 + \varsigma^2)}{\sqrt{T}} + \frac{n^3 \rho_{max}^4 L^2(\sigma^2 + \varsigma^2)}{T\rho_{min}\lambda_2^3}\right)$. We leave the proof of this last extension to the full version of this work.

## 5 Experimental Results

In this section, we validate our analysis, by applying the algorithm to training deep neural networks for image classification and machine translation. We map the algorithm onto a multi-node supercomputing setting, in which we have a large number of compute nodes, connected by fast communication links. The key overhead in this setting is *synchronization*: at large node counts, the cost of synchronizing all nodes so they execute in lock-step can be very high, see e.g. [Li et al., 2019] for numerical results on different workloads. Transmission cost also becomes significant at large node counts and large model sizes. Decentralized methods can mitigate this overhead, since nodes synchronize only sporadically and in pairs.

**Target System and Implementation.** We run SwarmSGD on the CSCS Piz Daint supercomputer, which is composed of Cray XC50 nodes, each with a Xeon E5-2690v3 CPU and an NVIDIA Tesla P100 GPU, using a state-of-the-art Aries interconnect over a Dragonfly network topology, which is *regular*. Please see Piz [2019] for more details. We implemented SwarmSGD in Pytorch and TensorFlow using MPI-based primitives, with non-blocking averaging. The Pytorch implementation is on top of SGP framework [Assran et al., 2018], and uses SwarmSGD to train ResNets on the CIFAR-

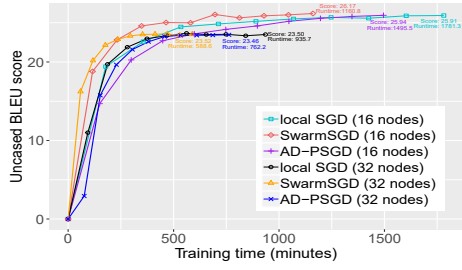

(a) Convergence versus Time.

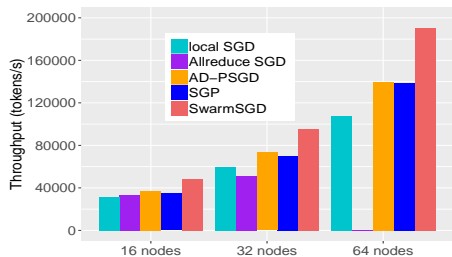

(b) Throughput comparison. Higher is better.

Figure 1: Convergence and Scalability on the Transformer/WMT Task with multiplier = 1.

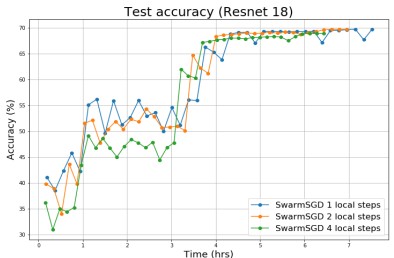

(a) Convergence in time versus number of local steps for ResNet18 on ImageNet. All variants recover the target accuracy, but we note the lower convergence of variants with more local steps. The experiment is run on 32 nodes.

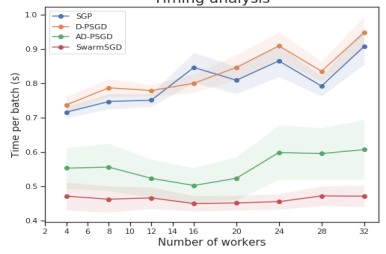

(b) Average time per batch for previous methods, compared to SwarmSGD, on ResNet18/ImageNet. The base value on the y axis (0.4) is the average computation time per batch, so values above represent average communication time per batch.

Figure 2: Convergence results and performance breakdown for ResNet18/ImageNet.

10/100 Krizhevsky et al. [2014] and ImageNet [Russakovsky et al., 2015] datasets, while we use the TensorFlow implementation to train the original version of the Transformer-XL model [Vaswani et al., 2017] on the WMT17 (En-Ge) dataset. All algorithms use the same topology overlay, which is *fully-connected*: according to their theory and experiments, this well-connected overlay should maximize convergence speed. SGP was run with overlap factor 1, following Assran et al. [2018].

**Training Process.** Our training methodology follows data-parallel training, with some differences due to decentralization, and is identical to previous work on decentralized and local SGD, e.g. Lian et al. [2017], Assran et al. [2018], Lin et al. [2018]. Training proceeds in *epochs*, each of which corresponds to processes collectively performing a full pass over the dataset. At the beginning of each epoch, we re-shuffle the dataset and partition it among processes [Lin et al., 2018].

As noted in previous work [Lian et al., 2017, 2018, Assran et al., 2018] variants of decentralized SGD are not always able to recover sequential SGD accuracy within the same number of epochs as this baseline. This is justified theoretically, as the slower mixing can affect convergence, but also intuitively, as each model sees significantly fewer updates per epoch. Thus, we will allow the decentralized schemes to execute for more epochs, by a constant *multiplier* factor between 1 and 3. To reduce multipliers, we experimented with SlowMo [Wang et al., 2019]; we found that it improved results across methods on CIFAR-10, but not at ImageNet scale; therefore, the provided results do not include it. Once we have fixed the number of epochs, *we do not alter the other training hyperparameters*: in particular, the learning rate schedule, momentum and weight decay terms are identical to the standard values for sequential SGD, for each individual model.

**Accuracy and Speed.** We first examined whether SwarmSGD can in fact recover full accuracy versus the sequential or large-batch SGD baselines. In Table 1 we provide an overview of parameter values to recover large-batch SGD accuracy (following Goyal et al. [2017]) using SwarmSGD, on the ResNet, ImageNet and CIFAR tasks. We execute for 32 nodes on ImageNet, and 8 nodes on CIFAR-10. (Local batch sizes are 128 for ResNet20 and ResNet50, and 128 for ResNet18. Quantization is not applied in these experiments.) The results show that Swarm can recover or slightly exceed the accuracy of the large-batch baselines, and that it has lower practical communication cost relative to existing methods (see Figure 2(b), where we separate the average computation cost per batch). However, Swarm requires significant additional passes over the data (up to 2.7×) to achieve

| Model / Dataset | SGD Top-1 | LB SGD Top-1 | SwarmSGD | Parameters |
|---|---|---|---|---|
| ResNet20 / CIFAR-10 | 91.7% (200 epochs) | 91.57% (200 epochs) | 91.59% (300 epochs) | 4 local steps |
| ResNet18 / ImageNet | 69.76 % (90 epochs) | 69.17% (90 epochs) | 69.79% (240 epochs) | 3 local steps |
| ResNet50 / ImageNet | 76.14% (90 epochs) | 76.01% (90 epochs) | 76.08% (240 epochs) | 2 local steps |

Table 1: Parameters for approximate Top-1 validation accuracy recovery on CIFAR-10 and ImageNet running on 32 nodes. Swarm step count represents *local SGD steps per model* between two averaging steps, and epochs are counted in terms of total passes over the data by all nodes.

full accuracy, which negates its performance benefits in this specific setting, relative to large-batch SGD. (Please see the Supplementary for end-to-end time comparisons.) This partly negative finding is in line with previous work on decentralized methods [Assran et al., 2018].

Next, we examine accuracy for the WMT17 task. The results are provided in Figure 1(a), in accuracy-vs-time format, for 16 and 32 nodes, executing for 10 global epochs. Here, the large-batch SGD (LB-SGD) baseline (BLEU score 26.1 at 16 nodes) is a poor alternative at high node counts due to model size: its throughput is low, and drops catastrophically at 64 nodes due to the network becoming severely bandwidth-bottlenecked (see Figure 1(b)). At 16 nodes, Swarm *slightly exceeds* the baseline accuracy at 26.17 BLEU, for an end-to-end speedup of $\sim 1.5\times$. In the same setting, Swarm outperforms all other decentralized methods (the fastest previous method, AD-PSGD, is 30% slower, and less accurate), both in terms of BLEU score, and in terms of end-to-end time.(The objective loss graph is similar, and is provided in the Appendix). At 32 nodes, all decentralized methods reach lower scores ($\sim 23.5$) after 10 epochs. However, we observed experimentally that running Swarm for an additional 5 epochs (multiplier 1.5) at 32 nodes recovered a BLEU score of $\sim 25.72$, which is 30% faster than the 16-node version in terms of end-to-end time (omitted for visibility).

In addition, we investigated 1) the accuracy of the *real average* of all models throughout training: it is usually more accurate than an arbitrary model, but not significantly so, corroborating the claim that individual models tend to stay close to the mean; 2) the influence of the number of local steps on accuracy: perhaps surprisingly, we were able to recover baseline accuracy on ResNet18/ImageNet for up to 4 local steps (see Figure 2(a)); 3) the impact of quantization on convergence, where we were able to recover accuracy when applying 8-bit model quantization to Swarm. We encourage the reader to examine the full experimental report in the Appendix, which contains data on these experiments, as well as additional ablation studies.

**Discussion.** Generally, the performance of SwarmSGD appears to be slighly superior to previous decentralized methods (see Figure 1 for an illustration, and Figure 2(b) for a performance breakdown). We investigated this advantage, and found that the per-step communication cost of Swarm, without quantization, is similar to AD-PSGD; however, our algorithm benefits from the *reduction in communication frequency*: nodes communicate at least 2x less often, and therefore incur lower *average* communication cost. In particular, a closer examination of the average batch times in Figure 2(b) shows that time per node per batch (including communication and computation) is largely *constant* as we increase the number of nodes, which suggests good scaling behaviour.

The main disadvantage of Swarm is that, similar to previous decentralized methods, it may need additional data passes in order to fully recover accuracy at high node counts. However, we also note that our method did not benefit from the high level of hyperparameter tuning applied to large-batch SGD, e.g. Goyal et al. [2017]. We find it interesting that this accuracy issue is less prevalent in the context of large, over-parameterized models, such as the Transformer, where Swarm can be a viable alternative to large-batch SGD within the same number of epochs.

## 6 Conclusions and Future Work

We analyzed the convergence of SGD in an extremely decoupled model of distributed computing, in which nodes mostly perform independent SGD updates, interspersed with intermittent pairwise averaging steps, which may be performed in an inconsistent and noisy manner. We showed that SGD still converges in this restrictive setting, even under consistency relaxations. Empirical results complement our analysis, showing that this method can outperform previous decentralized algorithms, and can even be competitive against large-batch SGD for very large models. A natural extension would be to generalize the bounds to arbitrary communication graphs. From the practical perspective, one extension would be to reduce the additional training epochs, and to experiment on large-scale decentralized testbeds.

## Acknowledgments and Disclosure of Funding

We gratefully acknowledge funding from the European Research Council (ERC) under the European Union's Horizon 2020 research and innovation programme (grant agreement No 805223 ScaleML). PD partly conducted this work while at IST Austria and was supported by the European Union's Horizon 2020 programme under the Marie Skłodowska-Curie grant agreement No. 754411. SL was funded in part by European Research Council (ERC) under the European Union's Horizon 2020 programme (grant agreement DAPP, No. 678880, and EPiGRAM-HS, No. 801039).

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
