# Contents

## A    Detailed Analytical Comparison

We compare our assumptions and the resulting bounds in more detail relative to Lian et al. [2018] , Assran et al. [2018] and Lu and De Sa [2020]. We focus on these works since they are the only other papers which do not require explicit global synchronization in the form of rounds. (By contrast, e.g. Wang and Joshi [2018], Koloskova et al. [2020] require that nodes synchronize in rounds, so that at every point in time every node has taken the same number of steps.)

### A.1    Comparison with SGP

In Assran et al. [2018], all nodes perform gradient steps at each iteration, in *synchronous* rounds, but averaging steps can be delayed by $\tau$ iterations. Unfortunately, the mixing time of their algorithm depends on the dimension $d$ (more precisely, it contains a $\sqrt{d}$ factor). Moreover, it depends on the delay bound $\tau$, and on $\Delta$, defined as the number of iterations over which the interaction graph is well connected. Additionally, their analysis will not extend to the random interaction patterns required by the asynchronous gossip models. Practically, their analysis works for deterministic global interactions, but where nodes may see inconsistent versions of eachothers' models. As noted in [Assran et al., 2018] and also in our Related Work section, enforcing the $\tau$ bound inherently implies that the algorithm may have to block in case a slow node may cause this bound to be violated.

### A.2    Comparison with AD-PSGD

Lian et al. [2018] consider random interaction matrices and do not require the agents to perform the same number of gradient steps. Unlike our model, in their case more than two nodes can interact during the averaging step. Due to asynchronous model, like ours, Lian et al. [2018] allow agents

to have outdated views during the averaging step. We would like to emphasize that in their case outdated models *are assumed to come from the same step*.

More precisely, at every step $t$, there exists $\tau \leq \tau_{max}$ such that for every agent $i$, $\hat{X}_t^i = X_{t-\tau}^i$. As also noted by Assran et al. [2018], enforcing this will require the usage of global barrier (or some alternate method of blocking while waiting for the nodes whose models are outdated by more than $\tau_{max}$ steps) once in every $\tau_{max}$ steps. Their implementation section suggests to explicitly implement synchronous pairings at every step.

In our case, at each step $t$ and agent $i$, the delay $\tau_t^i$ is a random variable , such that $t - \tau_i$ is the last step node $i$ was chosen as initiator. This implies naturally that $\hat{X}_t^i = X_{t-\tau}^i$, since $t - \tau_i$ is the last step node $i$ updated its own model.

### A.3 Comparison with Moniqua

Lu and De Sa [2020] consider a virtually identical model to AD-PSGD, but they also add quantization. The first difference between their work and ours is that we are using a random mixing matrix, thus we have to take the probability of models diverging (models diverge if the distance between them is larger then required by quantization algorithm) into account. Subsequently, this justifies our usage of Davies et al. [2021], since in this quantization method allows us to tolerate the larger distances between the models. This technical difference justifies the fact that our main bound has a non-trivial dependency on the second-moment bound $M$. As we showed, this dependency can be removed if we remove quantization. The second difference is that our interactions are one sided, that is, if $i$ and $j$ interact and $i$ is initiator, $i$ does not have to block while $j$ is in compute.

### A.4 Discussion

In summary, our algorithm is the first to explicitly consider the classic asynchronous gossip model [Xiao and Boyd, 2004], and show convergence bounds in its context. While AD-PSGD could be re-stated in this model, the corresponding bounds would be weaker than the ones we provide. At the practical level, to our knowledge we are the first to propose a fully non-blocking algorithm, which does not rely on a deterministic upper bound of $\tau_{\max}$ steps on the maximum delay between the nodes, and therefore remove the need for implementing global barrier-like communication to enforce $\tau_{\max}$. However, we note that our node activation model inherently implies a *probabilistic* bound on the expected maximum delay. In addition, we also allow for communication quantization and local steps, in the same asynchronous model.

The price we pay for this added generality is that the rate given in Theorem 4.1 has a dependency on the second-moment bound. As we showed in Corollary 4.2, this requirement can be removed if communication is not quantized.

## B The Complete Analysis

### B.1 Technical Lemmas on Load Balancing and Graph Properties

In this section provide the useful lemmas which will help as in the later sections.

We are given a simple undirected graph $G$, with $n$ nodes (for convenience we number them from 1 to $n$) and edge set $E(G)$. Let $\rho_i$ be a degree of vertex $i$ and let $\rho_i$ be a set of neighbours of $i$ ($|\rho_i| = \rho_i$). Also, we assume that the largest degree among nodes is $\rho_{max}$ and the smallest degree is $\rho_{min}$.

Each node $i$ of graph $G$ keeps a local vector model $X_t^i \in \mathbb{R}^d$ ($t$ is the number of interactions or steps); let $X_t = (X_t^1, X_t^2, ..., X_t^n)$ be the vector of local models at step $t$.

Let $\mu_t = \sum_{i=1}^n X_t^i / n$ be the average of models at step $t$ and let $\Gamma_t = \sum_{i=1}^n \|X_t^i - \mu_t\|^2$ be a potential at time step $t$.

Let $\mathcal{L}$ be the Laplacian matrix of $G$ and let let $\lambda_2$ be a second smallest eigenvalue of $\mathcal{L}$. For example, if $G$ is a complete graph $\lambda_2 = n$. In general we have that

$$\lambda_2 \leq 2\rho_{max}. \tag{5}$$

First we restate the following lemma from Ghosh and Muthukrishnan [1996]:

**Lemma B.1.**

$$\lambda_2 = \min_{v=(v_1, v_2, ..., v_n)} \left\{ \frac{v^T \mathcal{L} v}{v^T v} \Big| \sum_{i=1}^n v_i = 0 \right\} .$$

Now, we show that Lemma B.1 can be used to lower bound $\sum_{(i,j)\in E(G)} \|X_t^i - X_t^j\|^2$:

**Lemma B.2.**

$$\sum_{(i,j)\in E(G)} \|X_t^i - X_t^j\|^2 \geq \lambda_2 \sum_{i=1}^{n} \|X_t^i - \mu_t\|^2 = \lambda_2 \Gamma_t.$$

*Proof.* Observe that

$$\sum_{(i,j)\in E(G)} \|X_t^i - X_t^j\|^2 = \sum_{(i,j)\in E(G)} \|(X_t^i - \mu_t) - (X_t^j - \mu_t)\|^2. \tag{6}$$

Also, notice that Lemma B.1 means that for every vector $v = (v_1, v_2, ..., v_n)$ such that $\sum_{i=1}^{n} v_i = 0$, we have:

$$\sum_{(i,j)\in E(G)} (v_i - v_j)^2 \geq \lambda_2 \sum_{i=1}^{n} v_i^2.$$

Since $\sum_{i=1}^{n}(X_t^i - \mu_t)$ is a 0 vector, we can apply the above inequality to the each of $d$ components of the vectors $X_t^1 - \mu_t, X_t^2 - \mu_t, ..., X_t^n - \mu_t$ separately, and by elementary properties of 2-norm we prove the lemma.

Let $\rho_i$ be a degree of vertex $i$; we denote largest degree among nodes by $\rho_{max}$ and the smallest degree by $\rho_{min}$.

$\square$

## B.2   Properties of the Quantization Scheme

In this section, we discuss and prove the properties of the quantization scheme we consider, as well as potential complications caused by using quantization in an asynchronous shared-register setting. We also provide a fully-detailed version of Algorithm 3, with respect to quantization.

We first describe how the quantization method of Davies et al. [2021] is adapted to the local register setting to give Corollary 2.1:

## B.3   Proof of Corollary 2.1

**From Interactive Communication to Non-blocking Communication Buffers.** Lemma 23 of the full version of Davies et al. [2021] provides similar guarantees to Corollary 2.1, but in a different setting. Specifically, they assume interactive message-passing communication between an encoding node $u$ and a decoding node $v$. However, in their setting, the messages sent by $u$ are *non-adaptive*: $u$ simply sends quantizations using an increasing number of bits, until $v$ replies confirming that it has decoded successfully. In our setting, we implement communication buffers, so node $u$ can simply append all of its potential messages together as $Q_{R,\epsilon}(x_u)$.

Critically, notice that node $u$ should append enough bits so that the decoding is possible. This can be done in two ways. If $u$ knows the distance between $X_u$ and the vector $v$ uses for decoding, which we call the decoding key $key(v)$, then $u$ can simply write

$$O\left(d\log(\frac{R}{\epsilon}\|x_u - key(v)\|)\right) \text{ bits in the register.}$$

In the second case, $u$ does not know the distance. We can show, however, that with high probability in the total number of steps $T$, all distances between encoded and decoding vectors will be at most $\epsilon \cdot poly(T)$, and therefore at most $O(d\log T)$ bits for quantization will suffice in the worst case. Thus, the node writes $O(d\log T)$ bits in the register (Recall that $T$ is the total number of steps taken by our algorithm). But, when $v$ tries to decodes, it does not need all those bits: it reads and uses only the first $O\left(\log(\frac{R}{\epsilon}\|x_u - key(v)\|)\right)$ bits. This follows by Lemma 23 of Davies et al. [2021].

**Counting Communication Cost.** We emphasize that, when we calculate the number of bits used by quantization we actually aim to measure the number of bits *exchanged between* $u$ and $v$. In the setting we consider, which has local registers/communication buffers, this is the number of bits spent to read from (or to write to) the non-local register. Since the second case above involves writing a relatively large number of bits, we will use it only when $u$ is writing a quantized value to its *own* register/buffer, and so does not need to communicate the bits. Then, only the $O\left(\log(\frac{R}{\epsilon}\|x_u - key(v)\|)\right)$ bits read by $v$ need to be communicated.

To summarize, in our algorithm we will always ensure that whenever some node $u$ writes a quantized value, it either knows $key(v)$, or is writing to its local register. In the second case, we have to guarantee that $O(d \log T)$ bits suffice in the worst case. That is, $\|x_u - key(v)\| = O\left(\frac{\epsilon(poly(T))}{R}\right)$. In the first case, there are no restrictions.

## B.4 Notation and Auxiliary Potential Functions

Recall that $t - \tau_t^i$ is the last time $i$ was chosen as initiator up to and including step $t$. We would like to emphasize that $\tau_t^i$ is a random variable and we do not make any additional assumptions about it. Initially, $\tau_0^i = 0$ for every $i$. Then, if node $i$ is chosen as initiator at step $t + 1$ we have that

$$\tau_{t+1}^i = 0 \tag{7}$$

and for each

$$\tau_{t+1}^j = \tau_t^j + 1. \tag{8}$$

Next, we provide the formal definition of the local steps performed by our algorithms. Recall that $X_t^i$ is a local model of node $i$ at step $t$. Let $H_t^i$ be the number of local steps node $i$ performs in the case when it is chosen for interaction at step $t + 1$. A natural case is for $H_t^i$ to be fixed throughout the whole algorithm, that is: for each time step $t$ and node $i$, $H_t^i = H$ (or alternatively we might to try that $H_t^i$ can be a geometric r.v with mean $H$).

$$\widetilde{h}_i^0(X_t^i) = 0.$$

and for $1 \leq q \leq H_t^i$ let:

$$\widetilde{h}_i^q(X_t^i) = \widetilde{g}_i(X_t^i - \sum_{s=0}^{q-1} \eta \widetilde{h}_i^s(X_t^i)),$$

Note that stochastic gradient is recomputed at each step, but we omit the superscript for local step and global step for simplicity (Whenever we write $\widetilde{g}_i$, we mean that gradient is computed freshly by choosing sample u.a.r from the data available to node $i$). that is: $\widetilde{h}_i^{q,t}(X_t^i) = \widetilde{g}_i^{q,t}(X_t^i - \sum_{s=0}^{q-1} \eta \widetilde{h}_i^{s,t}(X_t^i))$. Further , for $1 \leq q \leq H_t^i$, let

$$h_i^q(X_t^i) = \mathbb{E}[\widetilde{g}_i(X_t^i - \sum_{s=0}^{q-1} \eta \widetilde{h}_i^s(X_t^i))] = \nabla f(X_t^i - \sum_{s=0}^{q-1} \eta \widetilde{h}_i^s(X_t^i))$$

be the expected value of $\widetilde{h}_i^q(X_t^i)$ taken over the randomness of the stochastic gradient $\widetilde{g}_i$. Let $\widetilde{h}_i(X_t^i)$ be the sum of $H_t^i$ local stochastic gradients we computed:

$$\widetilde{h}_i(X_t^i) = \sum_{q=1}^{H_t^i} \widetilde{h}_i^q(X_t^i).$$

In summary, omitting local step number $q$ means that we compute the sum of all generated gradients (this is the entire update during compute). and omitting tilde sign, means that we compute expectation over the randomness of the samples.

Similarly, for simplicity we avoid using index $t$ in the left side of the above definition, since it is clear that if the local steps are applied to model $X_t^i$ we compute them in the case when node $i$ is chosen as initiator at step $t + 1$.

In the case of outdated models this means that

$$\widetilde{h}_i(\hat{X}_t^i) = \widetilde{h}_i^{t-\tau_t^i}(X_{t-\tau_t^i})$$

**Potential Functions.** In order to deal with asynchrony we define the potential function: $\hat{\Gamma}_t = \sum_{i=1}^n \|\hat{X}_t^i - \mu_t\|^2$. This potential helps us to measure how far are the outdated models from the current average. In order to bound $\hat{\Gamma}_t$ in expectation, we will need additional auxiliary potential functions:

$$A_t = \sum_{i=1}^n \|X_{t-\tau_t i} - \mu_{t-\tau_t i}\|^2$$

$$B_t = \sum_{i=1}^n \|\mu_t - \mu_{t-\tau_t i}\|^2$$

Notice that by definition of $\hat{X}_t^i$ and Couchy-Schwarz inequality we get that

$$\hat{\Gamma}_t \leq 2A_t + 2B_t. \tag{9}$$

## B.5 Properties of Local Steps

**Lemma B.3.** *For any agent $i$ and step $t$*

$$\mathbb{E}\|\widetilde{h}_i(X_t^i)\|^2 \leq 2H^2M^2.$$

*Proof.*

$$\mathbb{E}\|\widetilde{h}_i(X_t^i)\|^2 = \sum_{K=1}^{\infty} Pr[H_t^i = K]\mathbb{E}\|\sum_{q=1}^{K} \widetilde{h}_i^q(X_t^i)\|^2$$

$$\overset{Cauchy-Schwarz}{\leq} \sum_{K=1}^{\infty} Pr[H_t^i = K]K\sum_{q=1}^{K} \mathbb{E}\|\widetilde{h}_i^q(X_t^i)\|^2$$

$$\overset{(??)}{\leq} \sum_{K=1}^{\infty} Pr[H_t^i = K]K^2M^2 \leq 2H^2M^2.$$

Where in the last step we used

$$\sum_{K=1}^{\infty} Pr[H_t^i = K]K^2 = \mathbb{E}[(H_t^i)^2] = 2H^2 - H \leq 2H^2.$$

$\square$

**Lemma B.4.** *For any agent $1 \leq i \leq n$, number of local steps $1 \leq K$ and step $t$, we have that*

$$\mathbb{E}\|\sum_{q=1}^{K} \widetilde{h}_i^q(\hat{X}_t^i)\|^2 \leq K\sigma^2 + 6L^2K\mathbb{E}\|\hat{X}_t^i - \mu_t\|^2 + \eta^2L^2K^2(K+1)(2K+1)M^2$$

$$+ 3K^2\mathbb{E}\|\nabla f_i(\mu_t) - \nabla f(\mu_t)\|^2 + 3K^2\mathbb{E}\|\nabla f(\mu_t)\|^2.$$

*Proof.*

$$\mathbb{E}\|\sum_{q=1}^{K} \widetilde{h}_i^q(\hat{X}_t^i)\|^2 \overset{(??)}{\leq} (K\sigma^2 + \mathbb{E}\|\sum_{q=1}^{K} h_i^q(\hat{X}_t^i)\|^2) = K\sigma^2 + \mathbb{E}\left\|\sum_{q=1}^{K} \nabla f_i(\hat{X}_t^i - \sum_{s=0}^{q-1} \eta\widetilde{h}_i^s(\hat{X}_t^i))\right\|^2$$

$$\overset{Cauchy-Schwarz}{\leq} K\sigma^2 + \sum_{q=1}^{K} K\mathbb{E}\left\|\left(\nabla f_i(\hat{X}_t^i - \sum_{s=0}^{q-1} \eta\widetilde{h}_i^s(\hat{X}_t^i)) - \nabla f_i(\mu_t)\right)\right.$$

$$\left.+ \nabla f_i(\mu_t) - \nabla f(\mu_t) + \nabla f(\mu_t)\right\|^2$$

$$\overset{Cauchy-Schwarz}{\leq} K\sigma^2 + 3K\sum_{q=1}^{K} \mathbb{E}\left\|\nabla f_i(\hat{X}_t^i - \sum_{s=0}^{q-1} \eta\widetilde{h}_i^s(\hat{X}_t^i)) - \nabla f_i(\mu_t)\right\|^2$$

$$+ 3K^2\mathbb{E}\|\nabla f_i(\mu_t) - \nabla f(\mu_t)\|^2 + 3K^2\mathbb{E}\|\nabla f(\mu_t)\|^2$$

$$\overset{Cauchy-Schwarz,(1)}{\leq} K\sigma^2 + 3L^2K\sum_{q=1}^{K} \mathbb{E}\left\|\hat{X}_t^i - \sum_{s=0}^{q-1} \eta\widetilde{h}_i^s(\hat{X}_t^i) - \mu_t\right\|^2$$

$$+ 3K^2\mathbb{E}\|\nabla f_i(\mu_t) - \nabla f(\mu_t)\|^2 + 3K^2\mathbb{E}\|\nabla f(\mu_t)\|^2$$

$$\overset{Cauchy-Schwarz}{\leq} K\sigma^2 + 6L^2K\mathbb{E}\|\hat{X}_t^i - \mu_t\|^2 + 6\eta^2L^2K\sum_{q=1}^{K} \mathbb{E}\left\|\sum_{s=0}^{q-1} \widetilde{h}_i^s(\hat{X}_t^i))\right\|^2$$

$$+ 3K^2\mathbb{E}\|\nabla f_i(\mu_t) - \nabla f(\mu_t)\|^2 + 3K^2\mathbb{E}\|\nabla f(\mu_t)\|^2$$

To finish the proof, we need to upper bound $\sum_{q=1}^{K} \mathbb{E} \left\| \sum_{s=0}^{q-1} \widetilde{h}_i^s(\hat{X}_t^i)) \right\|^2$:

$$\sum_{q=1}^{K} \mathbb{E} \left\| \sum_{s=0}^{q-1} \widetilde{h}_i^s(\hat{X}_t^i)) \right\|^2 \overset{Cauchy-Schwarz}{\leq} \sum_{q=1}^{K} q \left( \sum_{s=0}^{q-1} \mathbb{E} \left\| \widetilde{h}_i^s(\hat{X}_t^i)) \right\|^2 \right)$$

$$\overset{(4)}{\leq} \sum_{q=1}^{K} q^2 M^2 = K(K+1)(2K+1)M^2/6.$$

$\square$

Next, we sum up the upper bound given by the above lemma and take the randomness of the number local steps into the account:

**Lemma B.5.** *For any step t, we have that*

$$\sum_{i=1}^{n} \mathbb{E} \|\widetilde{h}_i(\hat{X}_t^i)\|^2 \leq nH\sigma^2 + 6L^2 H \mathbb{E}[\hat{\Gamma}_t] + 144n\eta^2 L^2 H^4 M^2 + 6nH^2 \varsigma^2 + 6nH^2 \mathbb{E}\|\nabla f(\mu_t)\|^2.$$

*Proof.* Using lemma B.4

$$\sum_{i=1}^{n} \mathbb{E} \|\widetilde{h}_i(\hat{X}_t^i)\|^2 = \sum_{i=1}^{n} \sum_{K=1}^{\infty} Pr[H_{t-\tau_t^i}^i = K] \mathbb{E} \| \sum_{q=1}^{K} \widetilde{h}_i^q(\hat{X}_t^i)\|^2$$

$$\leq \sum_{i=1}^{n} \sum_{K=1}^{\infty} Pr[H_{t-\tau_t^i}^i = K] \left( K\sigma^2 + 6L^2 K \mathbb{E}\|\hat{X}_t^i - \mu_t\|^2 \right.$$

$$+ \eta^2 L^2 K^2 (K+1)(2K+1)M^2$$

$$+ 3K^2 \mathbb{E}\|\nabla f_i(\mu_t) - \nabla f(\mu_t)\|^2$$

$$\left. + 3K^2 \mathbb{E}\|\nabla f(\mu_t)\|^2 \right)$$

Notice that $\sum_{K=1}^{\infty} Pr[H_{t-\tau_t^i}^i = K]K = H$, by the definition of expectation. Also,

$$\sum_{K=1}^{\infty} Pr[H_{t-\tau_t^i}^i = K]K^2 = \mathbb{E}[(H_t^i)^2] \leq 2H^2$$

and

$$\sum_{K=1}^{\infty} Pr[H_{t-\tau_t^i}^i = K]K^2(K+1)(2K+1) \leq 6 \sum_{K=1}^{\infty} Pr[H_{t-\tau_t^i}^i = K]K^4$$

$$= 6\mathbb{E}[(H_t^i)^4] \leq 144H^4.$$

Thus we get that:

$$\sum_{i=1}^{n} \mathbb{E}\|\widetilde{h}_i(\hat{X}_t^i)\|^2 \leq \sum_{i=1}^{n} \left( H\sigma^2 + 6L^2 K \mathbb{E}\|\hat{X}_t^i - \mu_t\|^2 + 36\eta^2 L^2 H^3 M^2 \right.$$

$$+ 6H^2 \mathbb{E}\|\nabla f_i(\mu_t) - \nabla f(\mu_t)\|^2$$

$$\left. + 6H^2 \mathbb{E}\|\nabla f(\mu_t)\|^2 \right)$$

$$\leq nH\sigma^2 + 6L^2 H \mathbb{E}[\hat{\Gamma}_t] + 144n\eta^2 L^2 H^4 M^2 + 6nH^2 \varsigma^2 + 6nH^2 \mathbb{E}\|\nabla f(\mu_t)\|^2.$$

Where in the last step we used the definition of $\hat{\Gamma}_t$ and (3). $\square$

**Lemma B.6.** *For any local step $1 \leq q$, and agent $1 \leq i \leq n$ and step t:*

$$\mathbb{E}\|\nabla f_i(\mu_t) - h_i^q(\hat{X}_t^i)\|^2 \leq 2L^2 \mathbb{E}\|\hat{X}_t^i - \mu_t\|^2 + 2L^2 \eta^2 q^2 M^2.$$

*Proof.*

$$\mathbb{E}\|\nabla f_i(\mu_t) - h_i^q(\hat{X}_t^i)\|^2 = \mathbb{E}\|\nabla f_i(\mu_t) - \nabla f_i(\hat{X}_t^i - \sum_{s=0}^{q-1} \eta\widetilde{h}_i^s(\hat{X}_t^i))\|^2$$

$$\overset{(1)}{\leq} L^2\mathbb{E}\|\mu_t - X_t^i + \sum_{s=0}^{q-1} \eta\widetilde{h}_i^s(\hat{X}_t^i))\|^2$$

$$\overset{Cauchy-Schwarz}{\leq} 2L^2\mathbb{E}\|\hat{X}_t^i - \mu_t\|^2 + 2L^2\eta^2\mathbb{E}\|\sum_{s=0}^{q-1} \widetilde{h}_i^s(\hat{X}_t^i)\|^2.$$

$$\overset{Cauchy-Schwarz}{\leq} 2L^2\mathbb{E}\|\hat{X}_t^i - \mu_t\|^2 + 2L^2\eta^2 q\sum_{s=0}^{q-1} \mathbb{E}\|\widetilde{h}_i^s(\hat{X}_t^i)\|^2$$

$$\overset{Cauchy-Schwarz}{\leq} 2L^2\mathbb{E}\|\hat{X}_t^i - \mu_t\|^2 + 2L^2\eta^2 q^2 M^2.$$

$\square$

**Lemma B.7.** *For any time step $t$.*

$$\sum_{i=1}^{n}\mathbb{E}\langle\nabla f(\mu_t), -h_i(\hat{X}_t^i)\rangle \leq 2HL^2\mathbb{E}[\hat{\Gamma}_t] - \frac{3Hn}{4}\mathbb{E}\|\nabla f(\mu_t)\|^2 + 12H^3nL^2M^2\eta^2.$$

*Proof.*

$$\sum_{i=1}^{n}\mathbb{E}\langle\nabla f(\mu_t), -h_i(\hat{X}_t^i)\rangle = \sum_{i=1}^{n}\sum_{K=1}^{\infty} Pr[H_{t-\tau_t^i}^i = K]\mathbb{E}\langle\nabla f(\mu_t), -\sum_{q=1}^{K} h_i^q(\hat{X}_t^i)\rangle$$

$$= \sum_{i=1}^{n}\sum_{K=1}^{\infty} Pr[H_{t-\tau_t^i}^i = K]\sum_{q=1}^{K}\Big(\mathbb{E}\langle\nabla f(\mu_t), \nabla f_i(\mu_t) - h_i^q(\hat{X}_t^i)\rangle - \mathbb{E}\langle\nabla f(\mu_t), \nabla f_i(\mu_t)\rangle\Big)$$

Using Young's inequality we can upper bound $\mathbb{E}\langle\nabla f(\mu_t), \nabla f_i(\mu_t) - h_i^q(\hat{X}_t^i)\rangle$ by

$$\frac{\mathbb{E}\|\nabla f(\mu_t)\|^2}{4} + \mathbb{E}\Big\|\nabla f_i(\mu_t) - h_i^q(\hat{X}_t^i)\Big\|^2.$$

Plugging this in the above inequality we get:

$$\sum_{i=1}^{n}\mathbb{E}\langle\nabla f(\mu_t), -h_i(\hat{X}_t^i)\rangle$$

$$\leq \sum_{i=1}^{n}\sum_{K=1}^{\infty} Pr[H_{t-\tau_t^i}^i = K]\sum_{q=1}^{K}\Big(\mathbb{E}\|\nabla f(\mu_t) - h_i^q(\hat{X}_t^i)\|^2$$

$$+ \frac{\mathbb{E}\|\nabla f(\mu_t)\|^2}{4} - \mathbb{E}\langle\nabla f(\mu_t), \nabla f_i(\mu_t)\rangle\Big)$$

$$\overset{\text{Lemma B.6}}{\leq} \sum_{i=1}^{n}\sum_{K=1}^{\infty} Pr[H_{t-\tau_t^i}^i = K]\sum_{q=1}^{K}\Big(2L^2\mathbb{E}\|\hat{X}_t^i - \mu_t\|^2 + 2L^2\eta^2 q^2 M^2$$

$$+ \frac{\mathbb{E}\|\nabla f(\mu_t)\|^2}{4} - \mathbb{E}\langle\nabla f(\mu_t), \nabla f_i(\mu_t)\rangle\Big)$$

$$= \sum_{i=1}^{n}\sum_{K=1}^{\infty} Pr[H_{t-\tau_t^i}^i = K]K\Big(2L^2\mathbb{E}\|\mu_t - \hat{X}_t^i\|^2 + \frac{\mathbb{E}\|\nabla f(\mu_t)\|^2}{4} - \mathbb{E}\langle\nabla f(\mu_t), \nabla f_i(\mu_t)\rangle\Big)$$

$$+ \sum_{i=1}^{n}\sum_{K=1}^{\infty} Pr[H_{t-\tau_t^i}^i = K]K(K+1)(2K+1)L^2M^2\eta^2/3$$

To finish the proof we upper bound the above two terms on the right hand side. Note that:

$$\sum_{i=1}^{n}\sum_{K=1}^{\infty} Pr[H_{t-\tau_t}^i = K]K\left(2L^2\mathbb{E}\|\mu_t - \hat{X}_t^i\|^2 + \frac{\mathbb{E}\|\nabla f(\mu_t)\|^2}{4} - \mathbb{E}\langle \nabla f(\mu_t), \nabla f_i(\mu_t)\rangle\right)$$

$$= \sum_{i=1}^{n} H\left(2L^2\mathbb{E}\|\mu_t - \hat{X}_t^i\|^2 + \frac{\mathbb{E}\|\nabla f(\mu_t)\|^2}{4} - \mathbb{E}\langle \nabla f(\mu_t), \nabla f_i(\mu_t)\rangle\right)$$

$$= H\left(2L^2\mathbb{E}[\hat{\Gamma}_t] - \frac{3n\mathbb{E}\|\nabla f(\mu_t)\|^2}{4}\right)$$

Where in the last step we used that $\sum_{i=1}^{n}\frac{f_i(x)}{n} = f(x)$, for any vector $x$. Also:

$$\sum_{i=1}^{n}\sum_{K=1}^{\infty} Pr[H_{t-\tau_t^i}^i = K]K(K+1)(2K+1)L^2M^2\eta^2/3$$

$$\leq \sum_{i=1}^{n}\sum_{u=1}^{\infty} Pr[H_{t-\tau_t}^i = K]2K^3L^2M^2\eta^2$$

$$\leq 12H^3nL^2M^2\eta^2.$$

Where in the last step we used (Recall that $H_{t-\tau_i}^i$ is a geometric random variable with mean $H$):

$$\sum_{K=1}^{\infty} Pr[H_{t-\tau_i}^i = K]K^3 = \mathbb{E}[(H_{t-\tau_i}^i)^3] \leq 6H^3.$$

$\square$

## B.6 Upper Bounding Potential Functions

We proceed by proving the following lemma which upper bounds the expected change in potential:

**Lemma B.8.** *For any time step $t$ we have:*

$$\mathbb{E}[\Gamma_{t+1}] \leq \left(1 - \frac{\lambda_2}{2n\rho_{max}}\right)\mathbb{E}[\Gamma_t] + \frac{20\rho_{max}^2}{\rho_{min}\lambda_2}(R^2+7)^2\epsilon^2 + \sum_{i}\frac{24\rho_{max}^2\eta^2}{\rho_{min}\lambda_2 n}\mathbb{E}\|\widetilde{h}_i(\hat{X}_t^i)\|^2.$$

*Proof.* First we bound change in potential $\Delta_t = \Gamma_{t+1} - \Gamma_t$ for some fixed time step $t > 0$.

For this, let $\Delta_t^{i,j}$ be the change in potential when agent $i$ wakes up (is chosen as initiator) and chooses neighbouring agent $j$ for interaction. Let $S_t^i = -\eta\widetilde{h}_i(\hat{X}_t^i) + \frac{Q(X_t^i)-X_t^i}{2} + \frac{Q(X_t^j)-X_t^j}{2}$ and $S_t^j = \frac{Q(X_t^i)-X_t^i}{2} + \frac{Q(X_t^j)-X_t^j}{2}$. We have that:

$$X_{t+1}^i = \frac{X_t^i + X_t^j}{2} + S_t^i.$$

$$X_{t+1}^j = \frac{X_t^i + X_t^j}{2} + S_t^j.$$

$$\mu_{t+1} = \mu_t + \frac{S_t^i + S_t^j}{n}.$$

This gives us that:

$$X_{t+1}^i - \mu_{t+1} = \frac{X_t^i + X_t^j}{2} + \frac{n-1}{n}S_t^i - \frac{1}{n}S_t^j - \mu_t.$$

$$X_{t+1}^i - \mu_{t+1} = \frac{X_t^i + X_t^j}{2} + \frac{n-1}{n}S_t^j - \frac{1}{n}S_t^i - \mu_t.$$

For $k \neq i, j$ we get that

$$X_{t+1}^k - \mu_{t+1} = X_t^k - \frac{1}{n}(S_t^i + S_t^j) - \mu_t.$$

Hence:

$$
\begin{aligned}
\Delta_t^{i,j} = {}& \left\| \frac{X_t^i + X_t^j}{2} + \frac{n-1}{n} S_t^i - \frac{1}{n} S_t^j - \mu_t \right\|^2 - \left\| X_t^i - \mu_t \right\|^2 \\
& + \left\| \frac{X_t^i + X_t^j}{2} + \frac{n-1}{n} S_t^j - \frac{1}{n} S_t^i - \mu_t \right\|^2 - \left\| X_t^j - \mu_t \right\|^2 \\
& + \sum_{k \neq i,j} \left( \left\| X_t^k - \frac{1}{n}(S_t^i + S_t^j) - \mu_t \right\|^2 - \left\| X_t^k - \mu_t \right\|^2 \right) \\
= {}& 2 \left\| \frac{X_t^i - \mu_t}{2} + \frac{X_t^j - \mu_t}{2} \right\|^2 - \left\| X_t^i - \mu_t \right\|^2 - \left\| X_t^j - \mu_t \right\|^2 \\
& + \left\langle X_t^i - \mu_t + X_t^j - \mu_t, \frac{n-2}{n} S_t^i + \frac{n-2}{n} S_t^j \right\rangle \\
& + \left\| \frac{n-1}{n} S_t^i - \frac{1}{n} S_t^j \right\|^2 + \left\| \frac{n-1}{n} S_t^j - \frac{1}{n} S_t^i \right\|^2 \\
& + \sum_{k \neq i,j} 2 \left\langle X_t^k - \mu_t, -\frac{1}{n}(S_t^i + S_t^j) \right\rangle \\
& + \sum_{k \neq i,j} (\frac{1}{n})^2 \| S_t^i + S_t^j \|^2 .
\end{aligned}
$$

Observe that:

$$
\sum_{k=1}^n \left\langle X_t^k - \mu_t, -\frac{1}{n}(S_t^i + S_t^j) \right\rangle = 0 .
$$

After combining the above two equations, we get that:

$$
\begin{aligned}
\Delta_t^{i,j} = {}& -\frac{\| X_t^i - X_t^j \|^2}{2} + \left\langle X_t^i - \mu_t + X_t^j - \mu_t, S_t^i + S_t^j \right\rangle \\
& + \frac{n-2}{n^2} \left\| S_t^i + S_t^j \right\|^2 + \left\| \frac{n-1}{n} S_t^i - \frac{1}{n} S_t^j \right\|^2 + \left\| \frac{n-1}{n} S_t^j - \frac{1}{n} S_t^i \right\|^2 \\
\overset{\text{Cauchy-Schwarz}}{\leq} {}& -\frac{\| X_t^i - X_t^j \|^2}{2} + \left\langle X_t^i - \mu_t + X_t^j - \mu_t, S_t^i + S_t^j \right\rangle \\
& + 2 \Big( \frac{n-2}{n^2} + \frac{1}{n^2} + \frac{(n-1)^2}{n^2} \Big) \Big( \| S_t^i \|^2 + \| S_t^j \|^2 \Big) \\
\leq {}& -\frac{\| X_t^i - X_t^j \|^2}{2} + \left\langle X_t^i - \mu_t + X_t^j - \mu_t, S_t^i + S_t^j \right\rangle \\
& + 2 \Big( \| S_t^i \|^2 + \| S_t^j \|^2 \Big) .
\end{aligned}
$$

Let $\alpha$ be a parameter we will fix later:

$$
\begin{aligned}
\left\langle X_t^i - \mu_t + X_t^j - \mu_t, S_t^i + S_t^j \right\rangle \overset{\text{Young}}{\leq} {}& \alpha \left\| X_t^i - \mu_t + X_t^j - \mu_t \right\|^2 + \frac{\left\| S_t^i + S_t^j \right\|^2}{4\alpha} \\
\overset{\text{Cauchy-Schwarz}}{\leq} {}& 2\alpha \left\| X_t^i - \mu_t \right\|^2 + 2\alpha \left\| X_t^j - \mu_t \right\|^2 + \frac{\left\| S_t^i \right\|^2 + \left\| S_t^j \right\|^2}{2\alpha} \\
\leq {}& 2\alpha \left\| X_t^i - \mu_t \right\|^2 + 2\alpha \left\| X_t^j - \mu_t \right\|^2 + \frac{\| S_t^i \|^2 + \| S_t^j \|^2}{2\alpha} .
\end{aligned}
$$

Finally, by combining the above two inequalities we get that

$$
\begin{aligned}
\Delta_t^{i,j} \leq {}& -\frac{\| X_t^i - X_t^j \|^2}{2} + \left\langle X_t^i - \mu_t + X_t^j - \mu_t, S_t^i + S_t^j \right\rangle \\
& + 2 \Big( \frac{n-2}{n^2} + \frac{1}{n^2} + \frac{(n-1)^2}{n^2} \Big) \Big( \| S_t^i \|^2 + \| S_t^j \|^2 \Big)
\end{aligned}
$$

$$\leq -\frac{\|X_t^i - X_t^j\|^2}{2} + 2\alpha \left\|X_t^i - \mu_t\right\|^2 + 2\alpha \left\|X_t^j - \mu_t\right\|^2$$
$$+ (2 + \frac{1}{2\alpha})\Big(\|S_t^i\|^2 + \|S_t^j\|^2\Big).$$

Using definitions of $S_t^i$ and $S_t^j$, Cauchy-Schwarz inequality and properties of quantization we get that

$$\|S_t^i\|^2 \leq 3\eta^2\|\widetilde{h}_i(\hat{X}_t^i)\|^2 + \frac{3\|Q(X_t^i) - X_t^i\|^2}{4} + \frac{3\|Q(X_t^j) - X_t^j\|^2}{4}$$
$$\leq 3\eta^2\|\widetilde{h}_i(\hat{X}_t^i)\|^2 + \frac{3(R^2 + 7)\epsilon^2}{2}.$$
$$\|S_t^j\|^2 \leq \frac{\|Q(X_t^i) - X_t^i\|^2}{2} + \frac{\|Q(X_t^j) - X_t^j\|^2}{2} \leq (R^2 + 7)\epsilon^2.$$

Next, we plug this in the previous inequality:

$$\Delta_t^{i,j} \leq -\frac{\|X_t^i - X_t^j\|^2}{2} + 2\alpha\left\|X_t^i - \mu_t\right\|^2 + 2\alpha\left\|X_t^j - \mu_t\right\|^2$$
$$+ (2 + \frac{1}{2\alpha})\Big(\frac{5(R^2+7)\epsilon^2}{2} + 3\eta^2\|\widetilde{h}_i(\hat{X}_t^i)\|^2\Big).$$

Next, we calculate probability of choosing edges from graph and upper bound $\Delta_t$ in expectation, for this we define $\mathbb{E}_t$ as expectation conditioned on the entire history up to and including step $t$

$$\mathbb{E}_t[\Delta_t] = \sum_i \sum_{j \in \rho_i} \frac{1}{n\rho_i}\mathbb{E}_t[\Delta_t^{i,j}]$$
$$\leq \sum_i \sum_{j \in \rho_i} \frac{1}{n\rho_i}\Big( -\frac{\|X_t^i - X_t^j\|^2}{2} + 2\alpha\left\|X_t^i - \mu_t\right\|^2 + 2\alpha\left\|X_t^j - \mu_t\right\|^2$$
$$+ (2 + \frac{1}{2\alpha})\Big(\frac{5(R^2+7)\epsilon^2}{2} + 3\eta^2\mathbb{E}_t\|\widetilde{h}_i(\hat{X}_t^i)\|^2\Big)\Big)$$
$$= -\sum_i \sum_{j \in \rho_i} \frac{\|X_t^i - X_t^j\|^2}{2n\rho_i}$$
$$+ \sum_i \frac{1}{n}(1 + \sum_{j \in \rho_i}\frac{1}{\rho_j})2\alpha\left\|X_t^i - \mu_t\right\|^2 + (5 + \frac{5}{4\alpha})(R^2+7)\epsilon^2$$
$$+ \sum_i \sum_{j \in \rho_i} \frac{1}{n\rho_i}(6 + \frac{3}{2\alpha})\eta^2\mathbb{E}_t\|\widetilde{h}_i(\hat{X}_t^i)\|^2$$

Now, we use the upper and lower bounds on the degree of vertices

$$\mathbb{E}_t[\Delta_t] \leq -\sum_i \sum_{j \in \rho_i} \frac{\|X_t^i - X_t^j\|^2}{2n\rho_{max}}$$
$$+ \sum_i \frac{1}{n}(1 + \sum_{j \in \rho_i}\frac{1}{\rho_{min}})2\alpha\left\|X_t^i - \mu_t\right\|^2 + (5 + \frac{5}{4\alpha})(R^2+7)\epsilon^2$$
$$+ \sum_i \sum_{j \in \rho_i} \frac{1}{n\rho_i}(6 + \frac{3}{2\alpha})\eta^2\mathbb{E}_t\|\widetilde{h}_i(\hat{X}_t^i)\|^2$$
$$\leq -\sum_{(i,j) \in E(G)} \frac{\|X_t^i - X_t^j\|^2}{n\rho_{max}}$$
$$+ \sum_j \frac{\rho_{max}}{\rho_{min}n}4\alpha\left\|X_t^j - \mu_t\right\|^2 + (5 + \frac{5}{4\alpha})(R^2+7)\epsilon^2$$

$$+ \sum_i \frac{1}{n}(6 + \frac{3}{2\alpha})\eta^2 \mathbb{E}_t \|\widetilde{h}_i(\hat{X}_t^i)\|^2$$

Now, we use lemma B.2:

$$\mathbb{E}_t[\Delta_t] \leq - \sum_{(i,j)\in E(G)} \frac{\lambda_2 \Gamma_t}{n\rho_{max}}$$

$$+ \frac{4\alpha \Gamma_t \rho_{max}}{\rho_{min} n} + (5 + \frac{5}{4\alpha})(R^2 + 7)\epsilon^2 + \sum_i \frac{1}{n}(6 + \frac{3}{2\alpha})\eta^2 \mathbb{E}_t \|\widetilde{h}_i(\hat{X}_t^i)\|^2.$$

By setting $\alpha = \frac{\lambda_2}{8\rho_{max}^2}$, we get that:

$$\mathbb{E}_t[\Delta_t] \leq - \frac{\lambda_2 \Gamma_t}{2n\rho_{max}}$$

$$+ (5 + 10\frac{\rho_{max}^2}{\rho_{min}\lambda_2})S_t^i + \sum_{i=1}^{n}(6 + 12\frac{\rho_{max}^2}{\rho_{min}\lambda_2})\eta^2 \mathbb{E}_t \|\widetilde{h}_i(\hat{X}_t^i)\|^2.$$

Next we remove the conditioning , and use the definitions of $\Delta_i$ and $S_t^i$ (for $S_t^i$ we also use upper bound which come from the properties of quantization).

$$\mathbb{E}[\mathbb{E}_t[\Gamma_{t+1}]] = \mathbb{E}[\Delta_t + \Gamma_t]$$

$$\leq \left(1 - \frac{\lambda_2}{2n\rho_{max}}\right)\mathbb{E}[\Gamma_t] + (5 + 10\frac{\rho_{max}^2}{\rho_{min}\lambda_2})(R^2 + 7)^2\epsilon^2$$

$$+ \sum_{i=1}^{n}(6 + 12\frac{\rho_{max}^2}{\rho_{min}\lambda_2})\eta^2 \mathbb{E}\|\widetilde{h}_i(\hat{X}_t^i)\|^2.$$

Finally, we get the proof of the lemma after using $\frac{\rho_{max}^2}{\rho_{min}\lambda_2} \geq \frac{1}{2}$ (See (5)) and regrouping terms. $\qquad\square$

This allows us to upper bound the potential in expectation for any step $t$.

**Lemma B.9.**

$$\mathbb{E}[\Gamma_t] \leq \frac{40n\rho_{max}^3}{\rho_{min}\lambda_2^2}(R^2 + 7)^2\epsilon^2 + \frac{96n\rho_{max}^3}{\rho_{min}\lambda_2^2}H^2M^2\eta^2.$$

*Proof.* We prove by using induction. Base case $t = 0$ trivially holds. For an induction step step we assume that $\mathbb{E}[\Gamma_t] \leq \frac{40n\rho_{max}^3}{\rho_{min}\lambda_2^2}(R^2 + 7)^2\epsilon^2 + \frac{96n\rho_{max}^3}{\rho_{min}\lambda_2^2}H^2M^2\eta^2$. We get that :

$$\mathbb{E}[\Gamma_{t+1}] \leq \left(1 - \frac{\lambda_2}{2n\rho_{max}}\right)\mathbb{E}[\Gamma_t] + \frac{20\rho_{max}^2}{\rho_{min}\lambda_2}(R^2 + 7)^2\epsilon^2 + \sum_i \frac{24\rho_{max}^2}{\rho_{min}\lambda_2 n}\mathbb{E}\|\widetilde{h}_i(\hat{X}_t^i)\|^2$$

$$\overset{\text{Lemma B.3}}{\leq} \left(1 - \frac{\lambda_2}{2n\rho_{max}}\right)\mathbb{E}[\Gamma_t] + \frac{20\rho_{max}^2}{\rho_{min}\lambda_2}(R^2 + 7)^2\epsilon^2 + \frac{48\rho_{max}^2\eta^2}{\rho_{min}\lambda_2}H^2M^2$$

$$\leq \left(1 - \frac{\lambda_2}{2n\rho_{max}}\right)\left(\frac{40\rho_{max}^3}{\rho_{min}\lambda_2^2}(R^2 + 7)^2\epsilon^2 + \frac{96\rho_{max}^3}{\rho_{min}\lambda_2^2}H^2M^2\eta^2\right)$$

$$+ \frac{20\rho_{max}^2}{\rho_{min}\lambda_2}(R^2 + 7)^2\epsilon^2 + \frac{48\rho_{max}^2\eta^2}{\rho_{min}\lambda_2}H^2M^2$$

$$= \frac{40n\rho_{max}^3}{\rho_{min}\lambda_2^2}(R^2 + 7)^2\epsilon^2 + \frac{96n\rho_{max}^3}{\rho_{min}\lambda_2^2}H^2M^2\eta^2.$$

$\qquad\square$

**Lemma B.10.** *For any time step* $t$:

$$\mathbb{E}[A_{t+1}] \leq (1 - \frac{1}{n})A_t + \frac{1}{n}\mathbb{E}[\Gamma_{t+1}].$$

*Proof.* Recall that if node $i$ is chosen as initiator at step $t$, then for each $j \neq i$,

$$\hat{X}_{t+1-\tau_{t+1}^j}^j - \mu_{t+1-\tau_{t+1}^j} = \hat{X}_{t-\tau_t^j}^j - \mu_{j-\tau_t^j},$$

since $\tau_{t+1}^j = \tau_t^j + 1$ and

$$\hat{X}_{t+1-\tau_{t+1}^i}^i - \mu_{t+1-\tau_{t+1}^i} = X_{t+1}^i - \mu_{t+1},$$

since $\tau_{t+1}^i = 0$ (See equations (7) and (8)). Thus, if $\mathbb{E}_t$ is expectation conditioned on the entire history up to and including step $t$ then

$$\mathbb{E}_t[A_{t+1} - A_t] = \sum_{i=1}^n \frac{1}{n}\left( \mathbb{E}_t\|X_{t+1}^i - \mu_{t+1}\|^2 - \|\hat{X}_{i-\tau_t^i}^i - \mu_{i-\tau_t^i}\| \right)$$

$$= \frac{1}{n}\mathbb{E}_t[\Gamma_{t+1}] - \frac{1}{n}A_t.$$

After removing conditioning and regrouping terms we get the proof of the lemma $\qquad\square$

Next, we upper bound $A_t$ in expectation

**Lemma B.11.** *For any time step $t$:*

$$\mathbb{E}[A_t] \le \frac{40 n \rho_{max}^3}{\rho_{min}\lambda_2^2}(R^2 + 7)^2 \epsilon^2 + \frac{96 n \rho_{max}^3}{\rho_{min}\lambda_2^2}H^2 M^2 \eta^2.$$

*Proof.* By combining Lemmas B.9 and B.10 we get that:

$$\mathbb{E}[A_{t+1}] \le (1 - \frac{1}{n})A_t + \frac{40\rho_{max}^3}{\rho_{min}\lambda_2^2}(R^2+7)^2\epsilon^2 + \frac{96\rho_{max}^3}{\rho_{min}\lambda_2^2}H^2 M^2 \eta^2$$

and the proof follows by using the same type of induction as in the proof of Lemma $B.9$

$\qquad\square$

Next we provide two different versions of upper bounding $\mathbb{E}\|\mu_{t+1} - \mu_t\|^2$, the first one will be useful for upper bounding $\mathbb{E}[B_t]$ and the second one will be used in the proof of convergence for SwarmSGD.

**Lemma B.12.** *For any time step $t$:*

$$\mathbb{E}\|\mu_{t+1} - \mu_t\|^2 \le \frac{6(R^2+7)^2\epsilon^2 + 6\eta^2 H^2 M^2}{n^2}.$$

*Proof.* Let $i$ be the agent which is chosen as initiator at step $t + 1$ and let $j$ be the neighbour it selected for interaction, also let $E_t$ be expectation which is condition on the entire history up to and including step $t$ We have that

$$\mathbb{E}_t\|\mu_{t+1} - \mu_t\|^2 = \frac{1}{n^2}\mathbb{E}_t\left\| Q(X_t^i) - X_t^i + Q(X_t^j - X_t^j - \eta\widetilde{h}_i(\hat{X}_t^i) \right\|^2$$

$$\overset{Cauchy-Schwarz}{\le} \frac{3}{n^2}\mathbb{E}_t\left\| Q(X_t^i) - X_t^i\right\| + \frac{3}{n^2}E_t\left\| Q(X_t^j) - X_t^j\right\|^2 + \frac{3\eta^2}{n^2}\mathbb{E}_t\left\|\widetilde{h}_i(\hat{X}_t^i)\right\|^2$$

$$\le \frac{6(R^2+7)^2\epsilon^2 + 6\eta^2 H^2 M^2}{n^2}.$$

Where in the last step we used property of quantization and Lemma B.3. Since this upper bound holds for any agents $i$ and $j$, after removing the conditioning, we get the proof of the lemma. $\quad\square$

**Lemma B.13.** *For any step $t$*

$$\mathbb{E}\|\mu_{t+1} - \mu_t\|^2 \le \frac{6(R^2+7)^2\epsilon^2}{n^2} + \frac{3\eta^2 H \sigma^2}{n^2} + \frac{18\eta^2 L^2 H \mathbb{E}[\hat{\Gamma}_t]}{n^3} + \frac{432\eta^4 L^2 H^4 M^2}{n^2}$$

$$+ \frac{18\eta^2 H^2 \varsigma^2}{n^2} + \frac{18\eta^2 H^2 \mathbb{E}\|\nabla f(\mu_t)\|^2}{n^2}.$$

*Proof.* Following the same steps as the proof of Lemma B.12 and taking the randomness of agents $i$ (the initiator) and $j$ interacting at step $t + 1$ in to the account we get that

$$\mathbb{E}\|\mu_{t+1} - \mu_t\|^2 \le \sum_{i=1}^n \sum_{j\in\rho_i} \frac{1}{n^3\rho_i}\left( 3\mathbb{E}\left\| Q(X_t^i) - X_t^i\right\| + 3\mathbb{E}\left\| Q(X_t^j) - X_t^j\right\|^2 \right.$$

$$+ 3\eta^2 \mathbb{E}\left\|\widetilde{h}_i(\hat{X}_t^i)\right\|^2\Bigg)$$

$$\leq \frac{6(R^2+7)^2\epsilon^2}{n^2} + \frac{3\eta^2}{n^3}\sum_{i=1}^n \mathbb{E}\|\widetilde{h}_i(\hat{X}_t^i)\|^2$$

$$\overset{\text{Lemma B.5}}{\leq} \frac{6(R^2+7)^2\epsilon^2}{n^2} + \frac{3\eta^2}{n^3}\Big(nH\sigma^2 + 6L^2H\mathbb{E}[\hat{\Gamma}_t] + 144n\eta^2L^2H^4M^2$$

$$+ 6nH^2\varsigma^2 + 6nH^2\mathbb{E}\|\nabla f(\mu_t)\|^2\Big)$$

$$= \frac{6(R^2+7)^2\epsilon^2}{n^2} + \frac{3\eta^2H\sigma^2}{n^2} + \frac{18\eta^2L^2H\mathbb{E}[\hat{\Gamma}_t]}{n^3} + \frac{432\eta^4L^2H^4M^2}{n^2}$$

$$+ \frac{18\eta^2H^2\varsigma^2}{n^2} + \frac{18\eta^2H^2\mathbb{E}\|\nabla f(\mu_t)\|^2}{n^2}.$$

$\square$

Our next goal is to upper bound $\mathbb{E}[B_t]$, for which we will need the following lemma:

**Lemma B.14.** *For any time step $t$ and agent $i$:*

$$\mathbb{E}\|\mu_t - \mu_{t-\tau_t^i}\|^2 \leq \frac{\mathbb{E}[(\tau_t^i)^2]\Big(6(R^2+7)^2\epsilon^2 + 6\eta^2H^2M^2\Big)}{n^2}.$$

*Proof.* Let $\mathbb{E}_{\tau_t^i}$ be an expectation which is conditioned on $\tau_t^i$

$$\mathbb{E}_{\tau_i}\|\mu_t - \mu_{t-\tau_t^i}\|^2 = \mathbb{E}_{\tau_t^i}\left\|\sum_{s=t-\tau_t^i}^{t-1}\left(\mu_{s+1} - \mu_s\right)\right\|^2 \overset{Cauchy-Schwarz}{\leq} \tau_t^i\sum_{s=t-\tau_t^i}^{t-1}\mathbb{E}_{\tau_t^i}\|\mu_{s+1} - \mu_s\|^2.$$

Note that for any $t - \tau_t^i \leq s \leq t-1$ we can use Lemma B.12 to upper bound $\mathbb{E}_{\tau_t^i}\|\mu_{s+1} - \mu_s\|^2$, since it uses quantization property and Lemma B.3 (which in turn uses (**??**) and the randomness of the number of local steps) which do not depend on $\tau_t^i$. In fact, as proof of Lemma $B.12$ suggests, for any $t - \tau_t^i \leq s \leq t-1$, we could condition $\mathbb{E}\|\mu_{s+1} - \mu_s\|^2$ on the entire history up to and including step $s$ (this history includes $\tau_t^i$ as well) and the upper bound would still hold. Thus, we get that

$$\mathbb{E}_{\tau_t^i}\|\mu_t - \mu_{t-\tau_t^i}\|^2 \leq \frac{(\tau_t^i)^2\Big(6(R^2+7)^2\epsilon^2 + 6\eta^2H^2M^2\Big)}{n^2}.$$

Finally we remove the conditioning :

$$\mathbb{E}\|\mu_t - \mu_{t-\tau_t^i}\|^2 = \mathbb{E}[\mathbb{E}_{\tau_t^i}\|\mu_t - \mu_{t-\tau_t^i}\|^2] \leq \frac{\mathbb{E}[(\tau_t^i)^2]\Big(6(R^2+7)^2\epsilon^2 + 6\eta^2H^2M^2\Big)}{n^2}.$$

Next, we proceed to prove the following lemma: $\square$

**Lemma B.15.** *for any step $t$*

$$\sum_{i=1}^n \mathbb{E}[(\tau_t^i)^2] \leq 5n^3. \tag{10}$$

*Proof.* For a fixed step $s$, let $\mathbb{E}_s$ be an expectation conditioned on the entire history up to and including step $s$. If agent $i$ is chosen as initiator at step $s+1$ then $\tau_{s+1}^i = 0$ and otherwise $\tau_{s+1}^i = \tau_s^i + 1$. Since $i$ is chosen with probability $\frac{1}{n}$, we have that

$$\sum_{i=1}^n \mathbb{E}_s[(\tau_{s+1}^i)^2] = \sum_{i=1}^n (1 - \frac{1}{n})\Big((\tau_s^i)^2 + 2\tau_s^i + 1\Big) \leq (1 - \frac{1}{n})\sum_{i=1}^n (\tau_s^i)^2 + 2\sum_{i=1}^n \tau_s^i + \frac{n^2}{2}.$$

Where in the last step we used that $n \geq 2$.

Also, by using Cauchy-Schwarz inequality we get that

$$\left(\sum_{i=1}^n \tau_s^i\right)^2 \leq n\left(\sum_{i=1}^n (\tau_s^i)^2\right) \iff \sum_{i=1}^n \tau_s^i \leq \sqrt{n\sum_{i=1}^n (\tau_s^i)^2}.$$

Thus,

$$\sum_{i=1}^{n} \mathbb{E}_s[(\tau_{s+1}^i)^2] \le (1 - \frac{1}{n}) \sum_{i=1}^{n} (\tau_s^i)^2 + 2\sqrt{n \sum_{i=1}^{n} (\tau_s^i)^2} + \frac{n^2}{2}.$$

Next, we remove the conditioning:

$$\sum_{i=1}^{n} \mathbb{E}[(\tau_{s+1}^i)^2] = \sum_{i=1}^{n} \mathbb{E}[\mathbb{E}_s[(\tau_{s+1}^i)^2]] \le (1 - \frac{1}{n}) \sum_{i=1}^{n} \mathbb{E}[(\tau_s^i)^2] + 2\mathbb{E}\sqrt{n \sum_{i=1}^{n} (\tau_s^i)^2} + \frac{n^2}{2}$$

$$\le (1 - \frac{1}{n}) \sum_{i=1}^{n} \mathbb{E}[(\tau_s^i)^2] + 2\sqrt{n \sum_{i=1}^{n} \mathbb{E}[(\tau_s^i)^2]} + \frac{n^2}{2}.$$

Where in the last step with use Jensen's inequality and concavity of square root function. Finally, we finish the proof of the lemma using induction. Base case holds trivially, for induction step we assume that $\sum_{i=1}^{n} \mathbb{E}[(\tau_{s+1}^i)^2] \le 5n^3$. We have that

$$\sum_{i=1}^{n} \mathbb{E}[(\tau_{s+1}^i)^2] \le (1 - \frac{1}{n}) \sum_{i=1}^{n} \mathbb{E}[(\tau_s^i)^2] + 2\sqrt{n \sum_{i=1}^{n} \mathbb{E}[(\tau_s^i)^2]} + \frac{n^2}{2}$$

$$\le (1 - \frac{1}{n})(5n^3) + 2\sqrt{5n^4} + \frac{n^2}{2} = 5n^3 + n^2(-5 + 2\sqrt{5} + \frac{1}{2}) \le 5n^3.$$

This finishes the proof of the Lemma. $\square$

Finally, we are ready to upper bound $B_t$

**Lemma B.16.** *For any step $t$:*

$$\mathbb{E}[B_t] \le 5n\Big(6(R^2 + 7)^2\epsilon^2 + 6\eta^2 H^2 M^2\Big).$$

*Proof.* Lemma B.14 gives us that

$$\mathbb{E}[B_t] = \sum_{i=1}^{n} \mathbb{E}\|\mu_t - \mu_{t-\tau_t^i}\|^2 \le \sum_{i=1}^{n} \frac{\mathbb{E}[(\tau_t^i)^2]\Big(6(R^2 + 7)^2\epsilon^2 + 6\eta^2 H^2 M^2\Big)}{n^2}$$

After applying Lemma B.15 we get the proof of the Lemma. $\square$

The last lemma in this section upper bounds $\mathbb{E}[\hat{\Gamma}_t]$:

**Lemma B.17.** *For any step $t$, we have that*

$$\mathbb{E}[\hat{\Gamma}_t] \le \frac{200n\rho_{max}^3}{\rho_{min}\lambda_2^2}(R^2 + 7)^2\epsilon^2 + \frac{312n\rho_{max}^3}{\rho_{min}\lambda_2^2} H^2 M^2 \eta^2$$

*Proof.* From (9), and Lemmas B.11 and B.16 we get that

$$\mathbb{E}[\hat{\Gamma}_t] \le 2\mathbb{E}[A_t] + 2\mathbb{E}[B_t] \le \frac{80n\rho_{max}^3}{\rho_{min}\lambda_2^2}(R^2 + 7)^2\epsilon^2 + \frac{192n\rho_{max}^3}{\rho_{min}\lambda_2^2} H^2 M^2 \eta^2$$

$$+ 5n\Big(6(R^2 + 7)^2\epsilon^2 + 6\eta^2 H^2 M^2\Big)$$

$$\le \frac{200n\rho_{max}^3}{\rho_{min}\lambda_2^2}(R^2 + 7)^2\epsilon^2 + \frac{312n\rho_{max}^3}{\rho_{min}\lambda_2^2} H^2 M^2 \eta^2$$

Where in the last step we used $\frac{\rho_{max}^2}{\rho_{min}\lambda_2} \ge \frac{1}{2}$ (See (5)). $\square$

### B.7 The Convergence of SwarmSGD

**Theorem B.18.** *For learning rate $\eta = n/\sqrt{T}$, Algorithm 1 converges at rate:*

$$\frac{1}{T} \sum_{t=0}^{T-1} \mathbb{E}\|\nabla f(\mu_t)\|^2 \le \frac{2(f(\mu_0) - f(x^*))}{H\sqrt{T}} + \frac{6(\sigma^2 + 6H\varsigma^2)}{\sqrt{T}}$$

$$+ \frac{1600\rho_{max}^3(R^2+7)^2\epsilon^2 L^2}{\rho_{min}\lambda_2^2} + \frac{2496n^2\rho_{max}^3 H^2 L^2 M^2}{T\rho_{min}\lambda_2^2}$$
$$+ \frac{78H^2 L^2 M^2 n^2}{T} + \frac{12(R^2+7)^2\epsilon^2\sqrt{T}}{Hn^2}.$$

*Proof.* Let $\mathbb{E}_t$ denote expectation conditioned on the entire history up to and including step $t$. By $L$-smoothness we have that

$$\mathbb{E}_t[f(\mu_{t+1})] \leq f(\mu_t) + \mathbb{E}_t\langle\nabla f(\mu_t), \mu_{t+1} - \mu_t\rangle + \frac{L}{2}\mathbb{E}_t\|\mu_{t+1} - \mu_t\|^2. \tag{11}$$

First we look at $\mathbb{E}_t\langle\nabla f(\mu_t), \mu_{t+1} - \mu_t\rangle = \langle\nabla f(\mu_t), \mathbb{E}_t[\mu_{t+1} - \mu_t]\rangle$. If agent $i$ is chosen as initiator at step $t+1$ and it picks its neighbour $j$ to interact, We have that

$$\mu_{t+1} - \mu_t = -\frac{\eta}{n}\widetilde{h}_i(\hat{X}_t^i) - (X_t^i - Q(X_t^i)) - (X_t^j - Q(X_t^j)).$$

Thus, in this case:

$$\mathbb{E}_t[\mu_{t+1} - \mu_t] = -\frac{\eta}{n}h_i(\hat{X}_t^i).$$

Where we used unbiasedness of quantization and stochastic gradients. We would like to note that even though we do condition on the entire history up to and including step $t$ and this includes conditioning on $\hat{X}_t^i$, the algorithm has not yet used $\widetilde{h}_i(\hat{X}_t^i)$ (it does not count towards computation of $\mu_t$), thus we can safely use all properties of stochastic gradients. Hence, we can proceed by taking into the account that each agent $i$ is chosen as initiator with probability $\frac{1}{n}$:

$$\mathbb{E}_t[\mu_{t+1} - \mu_t] = -\sum_{i=1}^n \frac{\eta}{n^2}h_i(\hat{X}_t^i).$$

and subsequently

$$\mathbb{E}_t\langle\nabla f(\mu_t), \mu_{t+1} - \mu_t\rangle = \sum_{i=1}^n \frac{\eta}{n^2}\mathbb{E}_t\langle\nabla f(\mu_t), -h_i(\hat{X}_t^i)\rangle.$$

Hence, we can rewrite (11) as:

$$\mathbb{E}_t[f(\mu_{t+1})] \leq f(\mu_t) + \sum_{i=1}^n \frac{\eta}{n^2}\mathbb{E}_t\langle\nabla f(\mu_t), -h_i(\hat{X}_t^i)\rangle + \frac{L}{2}\mathbb{E}_t\|\mu_{t+1} - \mu_t\|^2.$$

Next, we remove the conditioning

$$\mathbb{E}[(\mu_{t+1})] = \mathbb{E}[\mathbb{E}_t[f(\mu_{t+1})]] \leq \mathbb{E}[f(\mu_t)] + \sum_{i=1}^n \frac{\eta}{n^2}\mathbb{E}\langle\nabla f(\mu_t), -h_i(\hat{X}_t^i)\rangle + \frac{L}{2}\mathbb{E}\|\mu_{t+1} - \mu_t\|^2.$$

This allows us to use Lemmas B.13 and B.7:

$$\mathbb{E}[f(\mu_{t+1})] - \mathbb{E}[f(\mu_t)] \leq \frac{2\eta H L^2\mathbb{E}[\hat{\Gamma}_t]}{n^2} - \frac{3H\eta}{4n}\mathbb{E}\|\nabla f(\mu_t)\|^2 + \frac{12H^3 L^2 M^2\eta^3}{n}$$
$$+ \frac{6(R^2+7)^2\epsilon^2}{n^2} + \frac{3\eta^2 H\sigma^2}{n^2}$$
$$+ \frac{18\eta^2 L^2 H\mathbb{E}[\hat{\Gamma}_t]}{n^3} + \frac{432\eta^4 L^2 H^4 M^2}{n^2}$$
$$+ \frac{18\eta^2 H^2\varsigma^2}{n^2} + \frac{18\eta^2 H^2\mathbb{E}\|\nabla f(\mu_t)\|^2}{n^2}.$$

To simplify the above inequality we assume that $\eta \leq \frac{1}{8H}$ and also use the fact that $n \geq 2$. We get:

$$\mathbb{E}[f(\mu_{t+1})] - \mathbb{E}[f(\mu_t)] \leq \frac{4\eta H L^2\mathbb{E}[\hat{\Gamma}_t]}{n^2} - \frac{H\eta}{2n}\mathbb{E}\|\nabla f(\mu_t)\|^2 + \frac{39H^3 L^2 M^2\eta^3}{n}$$
$$+ \frac{6(R^2+7)^2\epsilon^2}{n^2} + \frac{3\eta^2 H(\sigma^2+6H\varsigma^2)}{n^2}.$$

Here, important thing is that we used $\frac{18\eta^2 H^2\mathbb{E}\|\nabla f(\mu_t)\|^2}{n^2} - \frac{H\eta\mathbb{E}\|\nabla f(\mu_t)\|^2}{4n} \leq 0$.

Further, we use Lemma B.17:

$$
\mathbb{E}[f(\mu_{t+1})] - \mathbb{E}[f(\mu_t)] \leq \frac{4\eta H L^2 \left( \frac{200 n \rho_{max}^3}{\rho_{min}\lambda_2^2}(R^2+7)^2\epsilon^2 + \frac{312 n \rho_{max}^3}{\rho_{min}\lambda_2^2}H^2 M^2 \eta^2 \right)}{n^2}
$$
$$
- \frac{H\eta}{2n}\mathbb{E}\|\nabla f(\mu_t)\|^2 + \frac{39 H^3 L^2 M^2 \eta^3}{n}
$$
$$
+ \frac{6(R^2+7)^2\epsilon^2 H L^2}{n^2} + \frac{3\eta^2 H(\sigma^2 + 6H\varsigma^2)}{n^2}
$$
$$
= \frac{800\eta\rho_{max}^3(R^2+7)^2\epsilon^2 H L^2}{n\rho_{min}\lambda_2^2} + \frac{1248\eta^3\rho_{max}^3 H^3 L^2 M^2}{n\rho_{min}\lambda_2^2}
$$
$$
- \frac{H\eta}{2n}\mathbb{E}\|\nabla f(\mu_t)\|^2 + \frac{39 H^3 L^2 M^2 \eta^3}{n}
$$
$$
+ \frac{6(R^2+7)^2\epsilon^2}{n^2} + \frac{3\eta^2 H(\sigma^2 + 6H\varsigma^2)}{n^2}.
$$

by summing the above inequality for $t = 0$ to $t = T - 1$, we get that

$$
\mathbb{E}[f(\mu_T)] - f(\mu_0) \leq -\sum_{t=0}^{T-1} \frac{\eta H}{2n}\mathbb{E}\|\nabla f(\mu_t)\|^2 + \frac{3\eta^2 H(\sigma^2 + 6H\varsigma^2)T}{n^2}
$$
$$
+ \frac{800\eta\rho_{max}^3(R^2+7)^2\epsilon^2 H L^2 T}{n\rho_{min}\lambda_2^2} + \frac{1248\eta^3\rho_{max}^3 H^3 L^2 M^2 T}{n\rho_{min}\lambda_2^2}
$$
$$
+ \frac{39 H^3 L^2 M^2 \eta^3 T}{n} + \frac{6(R^2+7)^2\epsilon^2 T}{n^2}
$$

Next, we regroup terms, multiply both sides by $\frac{2n}{\eta H T}$ and use the fact that $f(\mu_T) \geq f(x^*)$:

$$
\frac{1}{T}\sum_{t=0}^{T-1}\mathbb{E}\|\nabla f(\mu_t)\|^2 \leq \frac{2n(f(\mu_0) - f(x^*))}{H\eta T} + \frac{6\eta(\sigma^2 + 6H\varsigma^2)}{n}
$$
$$
+ \frac{1600\rho_{max}^3(R^2+7)^2\epsilon^2 L^2}{\rho_{min}\lambda_2^2} + \frac{2496\eta^2\rho_{max}^3 H^2 L^2 M^2}{\rho_{min}\lambda_2^2}
$$
$$
+ 78 H^2 L^2 M^2 \eta^2 + \frac{12(R^2+7)^2\epsilon^2}{n\eta H}
$$

Finally, we set $\eta = \frac{n}{\sqrt{T}}$:

$$
\frac{1}{T}\sum_{t=0}^{T-1}\mathbb{E}\|\nabla f(\mu_t)\|^2 \leq \frac{2(f(\mu_0) - f(x^*))}{H\sqrt{T}} + \frac{6(\sigma^2 + 6H\varsigma^2)}{\sqrt{T}}
$$
$$
+ \frac{1600\rho_{max}^3(R^2+7)^2\epsilon^2 L^2}{\rho_{min}\lambda_2^2} + \frac{2496 n^2 \rho_{max}^3 H^2 L^2 M^2}{T\rho_{min}\lambda_2^2}
$$
$$
+ \frac{78 H^2 L^2 M^2 n^2}{T} + \frac{12(R^2+7)^2\epsilon^2\sqrt{T}}{H n^2}. \tag{12}
$$

$\square$

**Proof of Corollary 4.2.** We get the proof by simply omitting quantization parameters $R$ and $\epsilon$ from the convergence bound given by the above theorem.

Our next goal is to show **how quantization affects the convergence**.

First we prove that the probability of quantization failing during the entire run of the algorithm is negligible.

**Lemma B.19.** *Let $T \geq 3n$, then for quantization parameters $R = 2 + T^{\frac{3}{d}}$ and $\epsilon = \frac{\eta H M}{(R^2 + 7)}$ we have that the probability of quantization never failing during the entire run of the Algorithm 1 is at least $1 - O\left(\frac{1}{T}\right)$.*

*Proof.* Let $\mathcal{L}_t$ be the event that quantization does not fail during step $t$. Our goal is to show that $Pr[\cup_{t=1}^T \mathcal{L}_t] \geq 1 - O\left(\frac{1}{T}\right)$. In order to do this, we first prove that $Pr[\neg \mathcal{L}_{t+1} | \mathcal{L}_1, \mathcal{L}_2, ..., \mathcal{L}_t] \leq O\left(\frac{1}{T^2}\right)$ (O is with respect to $T$ here).

Recall that up to this point we always assumed that quantization never fails, and we omitted conditioning on this event. Next, we rewrite our potential bounds but with the conditioning: Lemma B.9 gives us that for any step $t$

$$\mathbb{E}[\Gamma_t | \mathcal{L}_1, \mathcal{L}_2, ..., \mathcal{L}_t] \leq \frac{136 n \rho_{max}^3}{\rho_{min} \lambda_2^2} (R^2 + 7)^2 \epsilon^2. \tag{13}$$

and Lemma B.17 gives us that

$$\mathbb{E}[\hat{\Gamma}_t | \mathcal{L}_1, \mathcal{L}_2, ..., \mathcal{L}_t] \leq \frac{512 n \rho_{max}^3}{\rho_{min} \lambda_2^2} (R^2 + 7)^2 \epsilon^2 \tag{14}$$

Where we also used that $(R^2 + 7)\epsilon^2 = H^2 \eta^2 M^2$. We use this to upper bound the probability of failure due to the models being far away (in this case we will not be able to apply Corollary 2.1), for any fixed agent $i$ and its neighbour $j$. That is, we need need to lower bound probability that :

$$\|Q(\hat{X}_t^i) - X_t^i\|^2 \leq (R^{R^d} \epsilon)^2 \tag{15}$$

$$\|\hat{X}_t^i - \hat{X}_t^j\|^2 \leq (R^{R^d} \epsilon)^2 \tag{16}$$

$$\|\hat{X}_t^i - \hat{X}_t^j\|^2 = O\left(\frac{\epsilon^2 (poly(T))^2}{R^2}\right) \tag{17}$$

$$\|Q(\hat{X}_t^j) - X_t^j\|^2 \leq (R^{R^d} \epsilon)^2. \tag{18}$$

We would like to point out that these conditions are necessary for decoding to succeed, we ignore encoding since it will be counted when someone will try to decode it.

Notice that by using Cauchy-Schwarz we get that

$$\|Q(\hat{X}_t^i) - X_t^i\|^2 + \|\hat{X}_t^i - \hat{X}_t^j\|^2 + \|Q(\hat{X}_t^j) - X_t^j\|^2$$
$$\leq 3\|Q(\hat{X}_t^i) - \hat{X}_t^i\|^2 + 3\|\hat{X}_t^i - \mu_t\|^2 + 3\|\mu_t - X_t^i\|^2$$
$$+ 2\|\hat{X}_t^i - \mu_t\|^2 + 2\|\mu_t - \hat{X}_t^j\|^2$$
$$+ 3\|Q(\hat{X}_t^j) - \hat{X}_t^j\|^2 + 3\|\hat{X}_t^j - \mu_t\|^2 + 3\|\mu_t - X_t^j\|^2$$
$$\leq 10\hat{\Gamma}_t + 6\Gamma_t + 6(R^2 + 7)^2 \epsilon^2.$$

Since, $R = 2 + T^{\frac{3}{d}}$ this means that $(R^{R^d})^2 \geq 2^{2T^3} \geq T^{30}$, for large enough $T$. Hence, to lower bound probability that (15), (16), (17), (18) are be satisfied it is suffices to upper bound the probability that $10\hat{\Gamma}_t + 6\Gamma_t + 6(R^2 + 7)^2 \epsilon^2 \geq T^{30} \epsilon^2$:

For this, we use Markov's inequality:

$$Pr\left[(10\hat{\Gamma}_t + 6\Gamma_t + 6(R^2 + 7)^2 \epsilon^2) \geq T^{30} \epsilon^2 | \mathcal{L}_1, \mathcal{L}_2, ..., \mathcal{L}_t\right]$$
$$\leq \frac{\mathbb{E}[10\hat{\Gamma}_t + 6\Gamma_t + 6(R^2 + 7)^2 \epsilon^2 | \mathcal{L}_1, \mathcal{L}_2, ..., \mathcal{L}_t]}{T^{30} \epsilon^2}$$
$$\overset{(13),(14)}{\leq} \frac{\frac{5936 n \rho_{max}^3}{\rho_{min} \lambda_2^2} (R^2 + 7)^2 \epsilon^2 + 6(R^2 + 7)^2}{T^{30} \epsilon^2}$$
$$\leq O\left(\frac{1}{T^2}\right).$$

In the last step we used that $T \geq 3n$ and $\lambda_2 \geq \frac{1}{n^2}$ for a connected graph. Thus, the failure probability due to the models not being close enough for quantization to be applied is at most $O\left(\frac{1}{T^2}\right)$. Conditioned on the event that $\|Q(\hat{X}_t^i) - X_t^i\|$, $\|\hat{X}_t^i - \hat{X}_t^j\|$ and $\|Q(\hat{X}_t^j) - X_t^j\|$ are upper bounded by $T^{15} \epsilon$ (This is what we actually lower bounded the probability for using Markov), we get that the

probability of quantization algorithm failing is at most

$$\log\log(\frac{1}{\epsilon}\|Q(\hat{X}_t^i) - X_t^i\|) \cdot O(R^{-d})$$

$$+ \log\log(\frac{1}{\epsilon}\|\hat{X}_t^i - \hat{X}_t^j\|) \cdot O(R^{-d})$$

$$+ \log\log(\frac{1}{\epsilon}\|Q(\hat{X}_t^j) - X_t^j\|) \cdot O(R^{-d}) \leq O\left(\frac{\log\log T}{T^3}\right) \leq O\left(\frac{1}{T^2}\right).$$

Note that we do not need to union bound over all choices of $i$ and $j$, since we have just one inter-action and the above upper bound holds for any $i$ and $j$. By the law of total probability (to remove conditioning) and the union bound we get that the total probability of failure, either due to not being able to apply quantization or by failure of quantization algorithm itself is at most $O\left(\frac{1}{T^2}\right)$. Finally we use chain rule to get that

$$Pr[\cup_{t=1}^T \mathcal{L}_t] = \prod_{t=1}^T Pr[\mathcal{L}_t| \cup_{s=0}^{t-1} \mathcal{L}_s] = \prod_{t=1}^T \left(1 - Pr[\neg\mathcal{L}_t| \cup_{s=0}^{t-1} \mathcal{L}_s]\right)$$

$$\geq 1 - \sum_{t=1}^T Pr[\neg\mathcal{L}_t| \cup_{s=0}^{t-1} \mathcal{L}_s] \geq 1 - O\left(\frac{1}{T}\right).$$

In the end we would like to emphasize that we could get even better lower bound by scaling param-eter $R$ by constant factor. □

**Lemma B.20.** *Let $T \geq 3n$, then for quantization parameters $R = 2 + T^{\frac{3}{d}}$ and $\epsilon = \frac{\eta HM}{(R^2+7)}$ we have that the expected number of bits used by Algorithm 1 per step is $O\left(d\log\left(\frac{\rho_{max}^2}{\rho_{min}\lambda_2}\right)\right) + O\left(\log T\right).$*

*Proof.* If the initiator agent $i$ and its neighbour $j$ interact at step $t + 1$, Corollary 2.1 (Please see (**??**)) gives us that the total number of bits used is at most

$$O\left(d\log(\frac{R}{\epsilon}\|\hat{X}_t^i - \hat{X}_t^j\|)\right) + O\left(d\log(\frac{R}{\epsilon}\|Q(\hat{X}_t^j) - X_t^j\|)\right) + O\left(d\log(\frac{R}{\epsilon}\|Q(\hat{X}_t^j) - X_{t+1}^j\|)\right).$$

By taking the randomness of agent interaction at step $t+1$ into the account, we get that the expected number of bits used is at most:

$$\sum_{i=1}^n \sum_{j\in\rho_i} \frac{1}{n\rho_i}\left(O\left(d\log(\frac{R}{\epsilon}\|\hat{X}_t^i - \hat{X}_t^j\|)\right) + O\left(d\log(\frac{R}{\epsilon}\|Q(\hat{X}_t^j) - X_t^j\|)\right)\right.$$

$$\left. + O\left(d\log(\frac{R}{\epsilon}\|Q(\hat{X}_t^j) - X_{t+1}^j\|)\right)\right). \quad (19)$$

We proceed by upper bounding the first term:

$$\sum_{i=1}^n \sum_{j\in\rho_i} \frac{1}{n\rho_i}\left(O\left(d\log(\frac{R}{\epsilon}\|\hat{X}_t^i - \hat{X}_t^j\|)\right)\right) \leq \sum_{i=1}^n \sum_{j\in\rho_i} \frac{1}{n\rho_i}\left(O\left(d\log(\frac{R^2}{\epsilon^2}\|\hat{X}_t^i - \hat{X}_t^j\|^2)\right)\right)$$

$$\overset{Cauchy-Schwarz}{\leq} \sum_{i=1}^n \sum_{j\in\rho_i} \frac{1}{n\rho_i}\left(O\left(d\log\left(\frac{R^2}{\epsilon^2}(\|\hat{X}_t^i - \mu_t\|^2 + \|\hat{X}_t^j - \mu_t\|^2)\right)\right)\right)$$

$$\overset{Jensen}{\leq} O\left(d\log\left(\frac{R^2}{\epsilon^2}\sum_{i=1}^n \sum_{j\in\rho_i} \frac{1}{n\rho_i}(\|\hat{X}_t^i - \mu_t\|^2 + \|\hat{X}_t^j - \mu_t\|^2)\right)\right).$$

We have that

$$\sum_{i=1}^n \sum_{j\in\rho_i} \frac{1}{n\rho_i}(\|\hat{X}_t^i - \mu_t\|^2 + \|\hat{X}_t^j - \mu_t\|^2) = \sum_{i=1}^n \frac{1}{n}\|\hat{X}_t^i - \mu_t\|^2 + \sum_{i=1}^n \frac{1}{n}(\sum_{j\in\rho_i} \frac{1}{\rho_j})\|\hat{X}_t^j - \mu_t\|^2$$

$$\leq \sum_{i=1}^n \frac{1}{n}\|\hat{X}_t^i - \mu_t\|^2 + \sum_{j=1}^n \frac{\rho_{max}}{\rho_{min}n}\|\hat{X}_t^j - \mu_t\|^2$$

$$\leq \frac{2\hat{\Gamma}_t \rho_{max}}{\rho_{min} n}.$$

By combining this with the previous inequality we get that

$$\sum_{i=1}^{n} \sum_{j \in \rho_i} \frac{1}{n\rho_i} \left( O\Big(d\log(\frac{R}{\epsilon}\|\hat{X}_t^i - \hat{X}_t^j\|)\Big)\right) \leq O\left(d\log\Big(\frac{R^2 \rho_{max}}{\epsilon^2 \rho_{min}}(\frac{\hat{\Gamma}_t}{n})\Big)\right)$$

Next, notice that

$$O\Big(d\log(\frac{R}{\epsilon}\|Q(\hat{X}_t^j) - X_{t+1}^j\|)\Big) \leq O\Big(d\log(\frac{R^2}{\epsilon^2}\|Q(\hat{X}_t^j) - X_{t+1}^j\|^2)\Big)$$

$$\overset{Cauchy-Schwarz}{\leq} O\left(d\log\Big(\frac{R^2}{\epsilon^2}(\|Q(\hat{X}_t^j) - \hat{X}_t^j\|^2 + \|\hat{X}_t^j - \mu_t\|^2 \right.$$

$$\left. + \|\mu_t - \mu_{t+1}\|^2 + \|X_{t+1}^j - \mu_{t+1}\|^2)\Big)\right)$$

$$\leq O\left(d\log\Big(\frac{R^2}{\epsilon^2}((R^2+7)^2\epsilon^2 + \|\hat{X}_t^j - \mu_t\|^2 \right.$$

$$\left. + \|\mu_t - \mu_{t+1}\|^2 + \|X_{t+1}^j - \mu_{t+1}\|^2)\Big)\right)$$

Where in the last step we used Corollary 2.1. By following similar argument as above we can upper bound the third term in (19):

$$\sum_{i=1}^{n} \sum_{j \in \rho_i} \frac{1}{n\rho_i} \left( O\Big(d\log(\frac{R}{\epsilon}\|Q(\hat{X}_t^j) - X_{t+1}^j\|)\Big)\right)$$

$$\leq O\left(d\log\Big(\frac{R^2 \rho_{max}}{\epsilon^2 \rho_{min}}((R^2+7)^2\epsilon^2 + \|\mu_t - \mu_{t+1}\|^2 + \frac{\Gamma_{t+1}}{n} + \frac{\hat{\Gamma}_t}{n})\Big)\right)$$

Analogously, by using $Q(\hat{X}_t^j) - X_t^j = (Q(\hat{X}_t^j) - \hat{X}_t^j) + (\hat{X}_t^j - \mu_t) + (\mu_t - X_t^j)$ we can upper bound the second term in (19):

$$\sum_{i=1}^{n} \sum_{j \in \rho_i} \frac{1}{n\rho_i} \left( O\Big(d\log(\frac{R}{\epsilon}\|Q(\hat{X}_t^j) - X_t^j\|)\Big)\right)$$

$$\leq O\left(d\log\Big(\frac{R^2 \rho_{max}}{\epsilon^2 \rho_{min}}((R^2+7)^2\epsilon^2 + \frac{\Gamma_t}{n} + \frac{\hat{\Gamma}_t}{n})\Big)\right)$$

Hence, the expected number of bits used is at most

$$O\left(d\log\Big(\frac{R^2 \rho_{max}}{\epsilon^2 \rho_{min}}((R^2+7)^2\epsilon^2 + \|\mu_t - \mu_{t+1}\|^2 + \frac{\Gamma_{t+1}}{n} + \frac{\Gamma_t}{n} + \frac{\hat{\Gamma}_t}{n})\Big)\right),$$

since the above term is an upper bound for all the three terms in (19).

Next, we take the expectations of $\Gamma_t$, $\Gamma_{t+1}$, $\hat{\Gamma}_t$ and $\|\mu_t - \mu_{t+1}\|^2$ into the account. We get that the expected number of bits used is at most,

$$O\left(d\mathbb{E}\left[\log\Big(\frac{R^2 \rho_{max}}{\epsilon^2 \rho_{min}}((R^2+7)^2\epsilon^2 + \|\mu_t - \mu_{t+1}\|^2 + \frac{\Gamma_{t+1}}{n} + \frac{\Gamma_t}{n} + \frac{\hat{\Gamma}_t}{n})\Big)\right]\right)$$

$$\overset{Jensen}{\leq} O\left(d\log\Big(\frac{R^2 \rho_{max}}{\epsilon^2 \rho_{min}}((R^2+7)^2\epsilon^2 + \mathbb{E}\|\mu_t - \mu_{t+1}\|^2 + \frac{\mathbb{E}[\Gamma_{t+1}]}{n} + \frac{\mathbb{E}[\Gamma_t]}{n} + \frac{\mathbb{E}[\hat{\Gamma}_t]}{n})\Big)\right).$$

Notice that since $(R^2+7)^2\epsilon^2 = \eta^2 H^2 M^2$, Lemma B.9 gives us that both $\mathbb{E}[\Gamma_t]$ and $\mathbb{E}[\Gamma_{t+1}]$ are $O(\frac{\rho_{max}^3}{\rho_{min}\lambda_2^2}(R^2+7)^2\epsilon^2)$, Lemma B.17 gives us that $\mathbb{E}[\hat{\Gamma}_t] = O(\frac{\rho_{max}^3}{\rho_{min}\lambda_2^2}(R^2+7)^2\epsilon^2)$ as well and finally Lemma B.12 gives us that $\mathbb{E}\|\mu_t - \mu_{t+1}\|^2 = O(\frac{(R^2+7)\epsilon^2}{n^2})$. Thus, by plugging these upper

bounds in the above inequality we get that the expected number of bits used is at most

$$O\left(d\log\left(\frac{R^2\rho_{max}}{\epsilon^2\rho_{min}}((1+\frac{1}{n^2})(R^2+7)^2\epsilon^2+\frac{3\rho_{max}^3(R^2+7)^2\epsilon^2}{\rho_{min}\lambda_2^2})\right)\right)$$

$$=O\left(d\log\left(\frac{\rho_{max}^4(R^2+7)^2R^2}{\rho_{min}^2\lambda_2^2}\right)\right)$$

$$=O\left(d\log\left(\frac{\rho_{max}^2}{\rho_{min}\lambda_2}\right)\right)+O\left(d\log R\right)$$

$$=O\left(d\log\left(\frac{\rho_{max}^2}{\rho_{min}\lambda_2}\right)\right)+O\left(d\log(T^{3/d})\right)$$

$$=O\left(d\log\left(\frac{\rho_{max}^2}{\rho_{min}\lambda_2}\right)\right)+O\left(\log T\right).$$

$\square$

With this we can prove the main theorem:

**Theorem 4.1.** *For learning rate $\eta=n/\sqrt{T}$, where $T\geq 3n$ and quantization parameters $R=2+T^{\frac{3}{d}}$ and $\epsilon=\frac{\eta HM}{(R^2+7)}$, with probability at least $1-O(\frac{1}{T})$ we have that the Algorithm 1 converges at rate*

$$\frac{1}{T}\sum_{t=0}^{T-1}\mathbb{E}\|\nabla f(\mu_t)\|^2\leq\frac{2(f(\mu_0)-f(x^*))}{H\sqrt{T}}+\frac{6(\sigma^2+6H\varsigma^2)}{\sqrt{T}}$$

$$+\frac{1600\rho_{max}^3n^2H^2M^2}{T\rho_{min}\lambda_2^2}+\frac{2496n^2\rho_{max}^3H^2L^2M^2}{T\rho_{min}\lambda_2^2}$$

$$+\frac{78H^2L^2M^2n^2}{T}+\frac{12HM^2}{\sqrt{T}}.$$

*and uses $O\left(d\log\left(\frac{\rho_{max}^2}{\rho_{min}\lambda_2}\right)\right)+O\left(\log T\right)$ communication bits per step in expectation.*

*Proof.* The proof simply follows from using Lemmas B.19 and B.20, and plugging the value of $(R^2+7)\epsilon=\eta HM$ in Theorem B.18. $\square$

## C  Additional Experimental Results

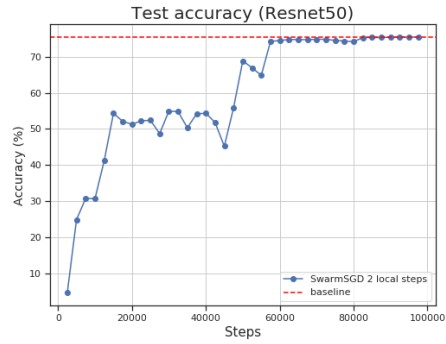

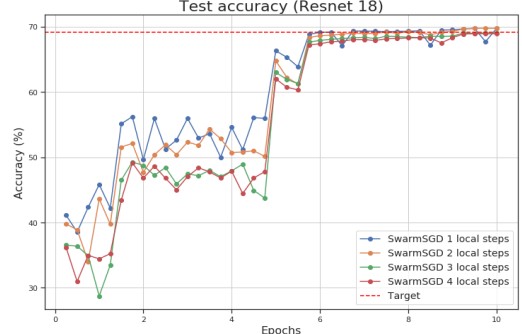

(a) Convergence of ResNet50/ImageNet versus number of gradient steps. SwarmSGD is able to recover the baseline top accuracy.

(b) Convergence versus number of local steps for ResNet18 on ImageNet. All variants recover the target accuracy, but we note the lower convergence of variants with more local steps.

Figure 3: Additional convergence results for ImageNet dataset.

**Target System and Implementation.** We run SwarmSGD on the CSCS Piz Daint supercomputer, which is composed of Cray XC50 nodes, each with a Xeon E5-2690v3 CPU and an NVIDIA Tesla

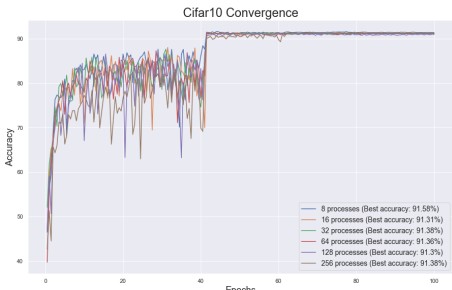

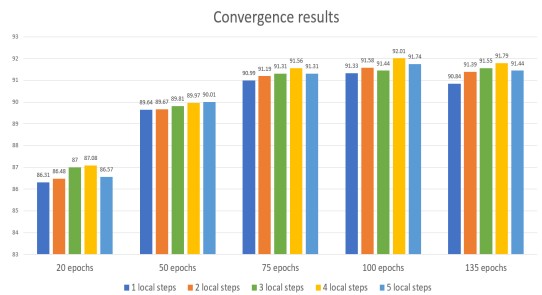

(a) Convergence versus number epochs (per model) for CIFAR-10/ResNet20, at node counts between 8 and 256. We note that the algorithm converges and recovers SGD accuracy (91.35% Top-1) for all node counts, although there are oscillations at high node counts.

(b) Accuracy versus local epochs and local steps for CIFAR-10/ResNet20. The original schedule for this model has 300 epochs, and this experiment is executed on 8 nodes. If the convergence scaling were perfect, $300/8 = 37.5$ epochs would have been sufficient to converge. However, in this case we need an epoch multiplier of 2, leading to 75 epochs.

Figure 4: Additional convergence results for CIFAR-10 dataset, versus number of nodes (left), and local steps (right).

P100 GPU, using a state-of-the-art Aries interconnect. Please see Piz [2019] for hardware details. We implemented SwarmSGD in Pytorch and TensorFlow using NCCL/MPI respectively. Basically, each node implements a computation thread, and a communication thread, each of which stores a copy of the model. The "live" copy, which is being updated with gradients, is stored by the computation thread. A simplified version of the Pytorch implementation is provided as additional material. When interacting, the two nodes exchange model information via their communication threads. Our implementation closely follows the non-blocking algorithm description.

We used SwarmSGD to train ResNets on the classic CIFAR-10/ImageNet datasets, and a Transformer Vaswani et al. [2017] on the WMT17 dataset (English-German).

**Hyperparameters.** The only additional hyperparameter is the total number of epochs we execute for. Once we have fixed the number of epochs, *we do not alter the other training hyperparameters*: in particular, the learning rate schedule, momentum and weight decay terms are identical to sequential SGD, for each individual model. Practically, if sequential SGD trains ResNet18 in 90 epochs, decreasing the learning rate at 30 and 60 epochs, then SwarmSGD with 32 nodes and multiplier 2 would $90 * 2/32 \simeq 5.6$ epochs per node, decreasing the learning rate at 2 and 4 epochs. As mentioned, we have also tried to use SlowMo [Wang et al., 2019], but did not observe significant improvements in terms of accuracy on ImageNet.

Specifically, for the ImageNet experiments, we used the following hyper-parameters. For ResNet18 and ResNet50, we ran for $240$ total parallel epochs using $32$ parallel nodes. The first communicated every 3 local steps, whereas the second communicated every 2 local steps. We used the same hyper-parameters (initial learning rate 0.1, annealed at 1/3 and 2/3 through training, and standard weight-decay and momentum parameters).

For the WMT17 experiments, we ran a standard Transformer-large model, and executed for $10$ *global* epochs at 16, 32, and 64 nodes. We ran a version with multiplier 1 (i.e. $10/\text{NUM\_NODES}$ epochs per model) and one with multiplier 1.5 (i.e. $15/\text{NUM\_NODES}$ epochs per model) and registered the BLEU score for each.

**Baselines.** We consider the following baselines:

- **Data-parallel SGD:** Here, we consider both the small-batch (strong scaling) version, which executes a global batch size of 256 on ImageNet/CIFAR experiments, and the large-batch (weak-scaling) baseline, which maximizes the batch per GPU. For the latter version, the learning rate is tuned following Goyal et al. [2017].

- **Local SGD [Stich, 2018, Lin et al., 2018]**: We follow the implementation of Lin et al. [2018], communicating globally every 5 SGD steps (which was the highest setting which provided good accuracy on the WMT task).

- **Previous decentralized proposals:** We experimented also with D-PSGD Lian et al. [2017], AD-PSGD Lian et al. [2018], and SGP Assran et al. [2018]. Due to computational constraints, we did

not always measure their end-to-end accuracy. Our method matches the sequential / large-batch accuracy for the models we consider within $1\%$. We note that the best performing alternative (AD-PSGD) is known to drop accuracy relative to the baselines, e.g. Assran et al. [2018].

Our Pytorch implementation builds upon that of Assran et al. [2018].

**Results.** The accuracy results for ImageNet experiments are given in Table 1 and Figures 3(a) and 3(b). As is standard, we follow Top-1 validation accuracy versus number of steps.

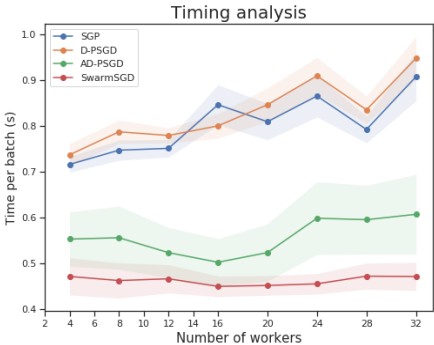

Figure 5: Average time per batch for previous methods, compared to SwarmSGD, on ResNet18/ImageNet, across 1000 repetitions with warm-up. Notice that 1) the time per batch of SwarmSGD stays *constant* relative to the number of nodes; 2) it is lower than any other method. This is due to the reduced communication frequency. Importantly, the base value on the y axis of this graph (0.4) is the average computation time per batch. Thus, everything above 0.4 represents the average communication time for this model. **We note that this comparison is performed in the framework of [Assran et al., 2018], and that we have considered the best-performing variant of D-PSGD, AD-PSGD and SGP, according to their implementation.**

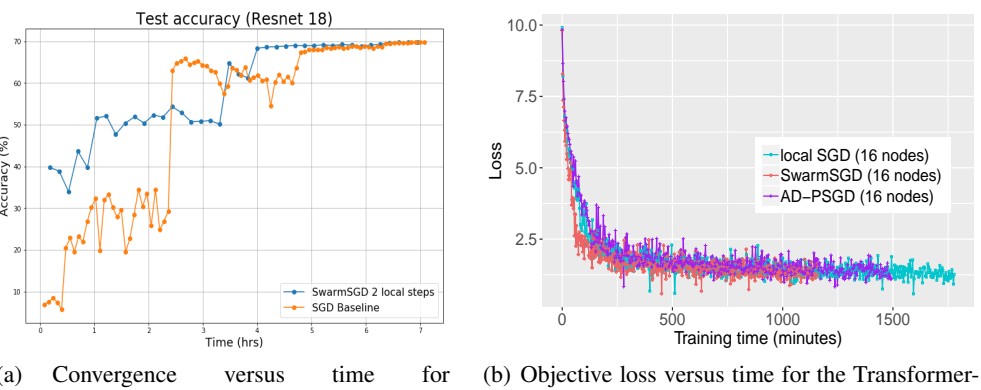

(a) Convergence versus time for ResNet18/Imagenet for the SGD baseline vs Swarm, executing at 32 nodes.

(b) Objective loss versus time for the Transformer-XL/WMT experiment, for various methods, executing at 16 nodes.

Figure 6: Convergence vs. time (ResNet18) and objective loss vs. time (Transformer).

**Communication cost.** We now look deeper into SwarmSGD's performance. For this, we examine in Figure 5 the average time per batch of different methods when executed on our testbed. The base value on the y axis (0.4s) is exactly the *average time per batch*, which is the same across all methods. Thus, the extra values on the y axis equate roughly to the communication cost of each algorithm. The results suggest that the communication cost can be up to half the cost of the full batch (for SGP and D-PSGD at large node counts). Moreover, this cost is *increasing* when considered relative to the number of workers (X axis), for all methods except SwarmSGD.

This reduced cost is justified simply because our method *reduces communication frequency:* it communicates less often, and therefore the average cost of communication at a step is lower. Figure 3(b) shows the convergence versus time for ResNet18 on the ImageNet dataset, at 32 nodes, with 3 local steps per node, and $\sim$ 7 epochs per model.

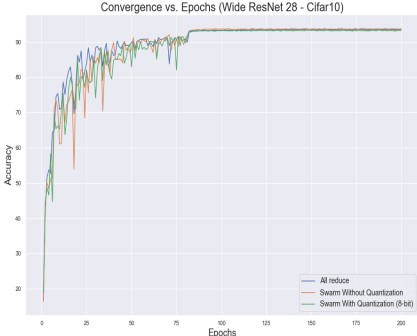
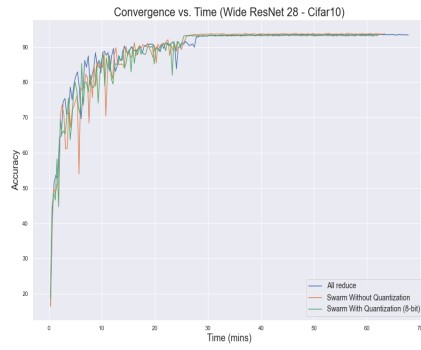

(a) Convergence versus number of steps for the quantized variant.

(b) Convergence versus time .

Figure 7: Convergence results for quantized 2xResNet28 trained on the CIFAR-10 dataset, versus iterations (left), and time (right).

**Convergence versus Steps and Epochs.** Figure 4 shows and discusses the results of additional ablation studies with respect to the number of nodes/processes and number of local steps / total epochs on the CIFAR-10 dataset / ResNet20 model. In brief, the results show that the method still preserves convergence even at very high node counts (256), and suggest a strong correlation between accuracy and the number of epochs executed per model. The number of local steps executed also impacts accuracy, but to a much lesser degree.

**Quantization.** Finally, we show convergence and speedup for a WideResNet-28 model with width factor 2, trained on the CIFAR-10 dataset. We note that the epoch multiplier factor in this setup is 1, i.e. Swarm (and its quantized variant) execute exactly the same number of epochs as the baseline. Notice that the quantized variant provides approximately $10\%$ speedup in this case, for a $< 0.3\%$ drop in Top-1 accuracy.