# OpenReview forum: "Asynchronous Decentralized SGD with Quantized and Local Updates"
_NeurIPS.cc/2021/Conference — NeurIPS 2021 Poster_

### Official Review · Reviewer_Zmf3 · 2021-06-29

**Rating:** 6
**Confidence:** 4

**Summary:**

This paper discusses a new variant for asynchronous decentralized SGD called SwarmSGD. Different from many previous algorithms SwarmSGD does not suffer from deadlocks. A new compression algorithm is used in SwarmSGD, which is different from commonly used ones such as stochastic rounding. Convergence rates of SwarmSGD are given with discussion on the assumption of second moment bound on the gradients, and experiments on language modeling and ImageNet are conducted.

**Limitations And Societal Impact:**

I don't see any potential negative societal impact of this paper.

**Main Review:**

The paper addresses an important question in asynchronous decentralized optimization, and is well written. The content is generally easy to follow.

Pros:
==
- The application of (Davies et al., 2021) in decentralized optimization seems interesting. The idea of compressing the distance of model rather than model itself is well-motivated.

- The paper provides detailed comparison to previous algorithms such as AD-PSGD and SGP (some of them are in the appendix), which clearly states the contribution and the challenges.

- I didn't check all the proof line by line, but the overall logic and the asymptotic complexity seems reasonable to me.

- The experiments are comparatively large-scale compared to other algorithm-type papers in the community.

Cons:
==
- In Theorem 4.1, the last term in the convergence rate is a little counter-intuitive. If SwarmSGD runs on a fully connected graph, it is expected to converge faster since the spectral gap increases. However, if we replace $\rho_{max}=n$ from complete graph into the last term, it becomes $O(n^5)$; and if in such case we need the $1/\sqrt{T}$ to be the leading term, we would require $T\geq O(n^{10})$. This seems a little diverged from reality, could you elaborate why using a better graph worsen the convergence rate?

- I'm a little confused by the statement of non-blocking and staleness in this paper. For starters, if worker $i$ is updating the communication buffer of worker $j$ while worker $j$ needs to read from its communication buffer for interacting with worker $m$, it should be waiting for $i$ to finish writing the buffer. Otherwise, it causes additional error like the Hogwild setting. A completely asynchronous execution should be one-sided writing such as ADDP algorithm in this paper: (https://arxiv.org/pdf/1901.08215.pdf). In fact, the line 4 in Algorithm 1 already implicitly assumes some sort of blocking: two workers cannot be sampled simultaneously and the updates in one iteration is completely atomic. Same thing applies to the staleness: if node $i$ is sampled uniformly, a staleness bound is implicitly placed.

- In the abstract the author states the algorithm works for heterogeneous data case, which does not seem to hold in the main theorem. Specifically, a bound of $\varsigma^2$ is still assumed in Equation (3) and shown in all the convergence rate. An algorithm robust to the heterogeneous case should be invariant to this bound. See the Theorem 2 of D$^2$: http://proceedings.mlr.press/v80/tang18a/tang18a.pdf.

-  A minor concern: In AD-PSGD, the $T$ is the number of query to gradient oracles while in this paper, $T$ is defined as the total number of interactions. Since each node is performing $H$ local steps. For fair comparison, the $T$ in convergence rate should be replaced by $TH$. However, this calibrated rate seems to have an additional $\sqrt{H}$ term on the sample complexity term $\sigma^2$. Why having local steps if it compromises the convergence rate while not having explicit benefits in the experiments?

**Updates**

I thank the authors for adequately addressing my concerns. I suggest including some of these points in the paper to make it clearer.


**Time Spent Reviewing:**

5

---

> ### Author Response · Authors · 2021-08-10
> **Response to Reviewer 4 (Zmf3)**
>
> > Convergence rate in a complete graph
>
> In the complete graph we have $ρ_{min}=n$ and the connectivity constant $\lambda_2=n$ as well. Thus, only $T \ge n^4$ suffices.
>
> > I'm a little confused by the statement of non-blocking and staleness in this paper. For starters, if worker $i$ is updating the communication buffer of worker j while worker $j$ needs to read from its communication buffer for interacting with worker $m$, it should be waiting for $i$ to finish writing the buffer. Otherwise, it causes additional error like the Hogwild setting.
>
> It is true that we assume that remote updates are atomic (instantaneous) once the update step has been applied, but we would argue that this assumption, and the fact that the buffers can never be in “partially written” state, are also used implicitly in all previous decentralized work, such as AD-PSGD or SGP. (Otherwise Hogwild!-style inconsistency would implicitly lead to dependencies in the dimension for all these papers, which is not the case.)
> Practically, our procedure can be implemented via primitives that are available in MPI implementations supporting RDMA one-sided reading and writing. Given this, the process is non-blocking with regards to e.g. the fact that the “receiving” node $X$ could be blocked forever during its computation step, even though its communication partners would still be able to make progress on its buffer.
> We would also note that this issue is purely theoretical: in the hundreds of runs we executed, we never observed blocking behaviour during the transfer step, as this would mean that the communication thread on a node crashed, which means that the node’s kernel is unstable.
>
> > A completely asynchronous execution should be one-sided writing such as ADDP algorithm in this paper: (https://arxiv.org/pdf/1901.08215.pdf).
>
> Thank you for this very interesting reference, which we will cover in detail in the next revision. The guarantees provided are indeed technically stronger than ours, as this work does not assume atomic remote updates. However, upon close inspection they also have non-trivial timing assumptions: namely, they assume lower and upper bounds on the frequency at which each node is activated during an interval, and, notably, they implicitly assume a bound on the time for each message to be delivered. (While not exactly comparable, these are similar in spirit to the assumptions we make.) We note that, in addition, the experimental setup is considerably simpler (also, there is no quantization or local steps).
> Relative to this reference, our work can be seen as partly non-blocking: in particular, nodes can be delayed arbitrarily long during their computation, which is the dominant component, without affecting global progress.
> We will add this reference to the discussion, and will clarify our assumptions and guarantees.
>
> > In fact, the line 4 in Algorithm 1 already implicitly assumes some sort of blocking: two workers cannot be sampled simultaneously and the updates in one iteration is completely atomic.
>
> We believe there is a misunderstanding here. As stated in the paper (and in the algorithm heading), we present Algorithm 1 from a _sequential_ perspective for simplicity first, in order to properly describe the quantization process. Then, in lines 243--263 we describe how this relates to the parallel execution. Specifically, in lines 256--263, we sketch what happens when a node is sampled concurrently by multiple interaction partners: their updates are ordered in a “communication queue,” which allows them to synchronize and obtain the correct update value to push.
> > Same thing applies to the staleness: if node i is sampled uniformly, a staleness bound is implicitly placed.
>
> Indeed, the staleness is in the order of $O(n \log{n})$ with high probability, due to the coupon collector bound. Notice however, that this is a probabilistic bound, and that the convergence bounds we get are considerably better than simply plugging this in as a simple staleness bound $\tau$ (which wouldn’t be formally correct). Having some kind of staleness bound assumption does appear to be inherent; this is the case even for the ADDP algorithm.
>
> > heterogeneous data
>
> We apologise for the confusion, by heterogeneous data we meant that nodes may optimize different local loss functions, but we still require a global variance bound. Our setup is similar to e.g. AD-PSGD, and matches our practical setting, where data is split randomly. Unlike the $D^2$ paper, we do require this additional assumption: This is precisely because our setting is asynchronous, with random activation times, whereas they consider a synchronous setting.
>
> > A minor concern: In AD-PSGD, the $T$ is the number of query to gradient oracles while in this paper, $T$ is defined as the total number of interactions. Since each node is performing $H$ local steps. For fair comparison, the $T$ in convergence rate should be replaced by $TH$. However, this calibrated rate seems to have an additional H term on the sample complexity term $\sigma^2$. Why having local steps if it compromises the convergence rate while not having explicit benefits in the experiments?
>
> The reviewer is correct: if we count gradient evaluations, then convergence should be counted in terms of $T*H$. (We also mention this in lines 274-275.) We chose to count “major” iterations, corresponding to the communication steps, since these represent the key cost of the algorithm.
> We will further clarify this and provide both measures.
> We stress that in practice we view H to be a small constant (it is 2-5 in our experiments), so this does not change the asymptotics in any way. Having $H > 1$ predictably lowers the convergence rate due to higher “variance” (which is true even in synchronous models [Koloskova et al.] and [Wang and Joshi]).
> However, even $H = 2$ leads to a significant improvement in practice, as it halves the frequency of synchronization. To illustrate, our implementation’s average per-step cost for $H = 1$ and $N = 4$ in Figure 2(b) is around 0.6--therefore slightly higher than AD-PSGD; setting $H = 2$ lowers it to 0.47 (as depicted). Thus, having local steps is definitely worth it in practice.

---

### Official Review · Reviewer_GmDi · 2021-07-16

**Rating:** 6
**Confidence:** 4

**Summary:**

This paper proposes a decentralized optimization method, SwarmSGD, for reducing the communication overhead in distributed deep learning. The proposed method combines asynchronous pair-wise gossip, quantized communication, and local updates into a single method and a unified analysis. Convergence guarantees are provided for smooth non-convex function, using the quantizer from Davies et al., 2021, and a bounded second moment assumption. Numerical results are provided on image classification (CIFAR, ImageNet) and machine translation (WMT En-Ge) tasks.

**Limitations And Societal Impact:**

There don't seem to be any obvious negative societal implications of this work.

**Main Review:**

### Originality and Significance
This paper mostly combines many related ideas from the literature (asynchronous pair-wise gossip, quantized communication, local updates) into a single method and so originality is limited. However, there is significance in the work since it is necessary for all these orthogonal communication reduction techniques to eventually be integrated together. In this sense, this paper fits nicely in the literature.

### Quality
I only have a couple of concerns about the quality of the empirical and theoretical contribution, and would appreciate if the authors could address these in their rebuttal.

The assumptions made consist of
- Static communication graph
- L-smooth
- Bounded variance
- Bounded second moment
- Poisson point process for asynchronous activation of nodes

To the best of my knowledge, the analysis applies standard tools from the literature, e.g., the supermartingale-esque iteration, used heavily in the analysis of subgradient push methods, as well as perron-frobenius for consensus. However, in my opinion the bounded second momentum assumption is a non-trivial limitation of the theoretical contribution. It may be ok in a quantized setting, but this assumption is still used in Corollary 4.2 after removing the quantization effects.

The Poisson point process assumption is *fine* for the theoretical assumptions, but then it is approximated in practice by setting the number of local gradient steps for each node to be a geometric random variable, which I would expect to impractically degrade performance in practice, since more frequent asynchronous non-blocking communication is likely to result in better convergence.

The baseline numbers for SwarmSGD seem good and consistent with the literature. Specifically, the proposed method looks to recover standard numbers for the chosen benchmarks. However the timing comparison with other gossip-based methods looks highly misleading. For example, it seems odd to use a fully-connected topology for the comparison of general gossip-based methods like SGP and D-PSGD with the pair-wise methods like SwarmSGD. Especially since the whole motivation for gossip-based deep learning methods is to reduce communication overhead by leveraging sparse topologies.

**Time Spent Reviewing:**

3

---

> ### Author Response · Authors · 2021-08-10
> **Response to Reviewer 3 (GmDi)**
>
> > This paper mostly combines many related ideas from the literature (asynchronous pair-wise gossip, quantized communication, local updates) into a single method and so originality is limited.
>
> The reviewer’s description is accurate; we would only want to point out that our quantization mechanism is fairly novel relative to the existing decentralized literature, and that its analysis technically non-trivial.
>
> > In my opinion the bounded second momentum assumption is a non-trivial limitation of the theoretical contribution. It may be ok in a quantized setting, but this assumption is still used in Corollary 4.2 after removing the quantization effects.
>
> We acknowledge this point, and would like to add two clarifications:
>
> In the context of Corollary 4.2, the second-moment term is divided by $T$, whereas all other terms are divided by $\sqrt{T}$. In practice, this would render that term asymptotically negligible. An identical assumption and argument are made in [Lu and De Sa, 2020], although their algorithm and proof are different.
>
>
> The second-moment assumption in Corollary 4.2 is only needed because of quantization and also because, we assume the number of local steps $H$ to be a random variable. In the absence of these factors (if $H$ is a fixed constant and there is no quantization), we can completely remove $M$ (which could be $\infty$), and only work with $\sigma$. We debated adding this extra corollary to the submission but were unable to due to space constraints. Instead we chose to show that without quantization, the contribution of terms containing $M$ is not significant. We would be happy to outline the proof during the discussion.
>
> > I would expect [geometric delays] to impractically degrade performance in practice, since more frequent asynchronous non-blocking communication is likely to result in better convergence.
>
> We agree on this point; however, we have noticed that in practice removing this assumption and fixing H to be constant does not significantly improve accuracy or performance. It would be simpler to assume that the batch running times come from an exponential distribution (and then the intercommunication times would also be exponential) but we do not have data to fully support such an assumption.
>
> > The baseline numbers for SwarmSGD seem good and consistent with the literature. Specifically, the proposed method looks to recover standard numbers for the chosen benchmarks. However the timing comparison with other gossip-based methods looks highly misleading. For example, it seems odd to use a fully-connected topology for the comparison of general gossip-based methods like SGP and D-PSGD with the pair-wise methods like SwarmSGD. Especially since the whole motivation for gossip-based deep learning methods is to reduce communication overhead by leveraging sparse topologies.
>
> We believe that there may be an unfortunate misunderstanding here, relative to our use of the term “topology.”
>
> Specifically, the reviewer may have understood that we run SGP and AD-PSGD with a fully-connected interaction topology, i.e. having each node interact with all other nodes in a step, akin to allreduce. However, by “topology” we mean that we allow any pair of nodes to interact in a “round,” but otherwise we run the algorithms in their standard form, i.e. using pairwise gossip-like updates.
> To be more precise, in SGP each node has a list of neighbors it cycles through, and in each step each node contacts the next single neighbor on its list. In AD-PSGD, we construct a random matching in each “step” by which nodes interact, but the matching is done in pairs.
>
> We chose to use a fully-connected underlying topology for two reasons:
> 1: In our practical setting (CSCS Piz Daint), the underlying network topology is extremely dense (most nodes are within one hop and all nodes were within two hops); artificially restricting it to a sparser topology, e.g. a ring, would not improve performance and would result in sub-optimal utilization of the hardware. (This may not be the case for e.g. AD-PSGD, which executed on a less performant network.)
> 2: We noticed that sparser topologies slow down convergence per SGD step in practice, and do not improve practical performance (see above). This is the case for both AD-PSGD, and is in keeping with its theoretical guarantees.
>
> We really hope this clarifies this last point; we would be happy to address this in more detail in the discussion. We will also add a note to this in the next version.

---

> > ### Comment · Reviewer_GmDi · 2021-09-02
> > **Response to Authors**
> >
> > I have read the author response and the other reviews, here are my thoughts:
> >
> > I appreciate the authors clarification about the quantizer, and still find the novelty limited, but I do think it is important for a work to actually take this step of combining these orthogonal approaches into a single method, and a unified analysis.
> >
> > On theory:
> > It seems a consistent point raised in the reviews that assumptions needed for the theory may be somewhat impractical/misleading for the proposed setting, such as the atomic operations for asynchronous gossip. Of course, without allowing workers to write directly to each other, the method would be susceptible to deadlock.
> > The bounded second moment assumption is a strong limitation, especially that it is also used in the non-quantized results. However, despite these limitations, I still see value in the theoretical contribution, and wouldn't hold these points against this work.
> >
> > On experiments:
> > Their proposed method seems to work well in practice, although I still find it a little strange to artificially activate nodes according to a poisson point process in practice.
> > There are questions on the empirical analysis raised by all reviewers, but the authors have clarified much of their experimental setup, and this is fine with me.
> >
> > In light of these points, I will probably stick to my weak accept score.

---

> > > ### Author Response · Authors · 2021-09-02
> > > **thank you + two minor clarifications**
> > >
> > > Thank you very much for your detailed response! We believe we have converged on most points, and will just make two very minor clarifications:
> > >
> > > 1. The second-moment bound requirement can in fact be completely removed, if we do not use quantization. We will state the resulting bound in the next version of our work.
> > >
> > > 2. We do not exactly activate nodes on a Poisson clock (which would be indeed a bit unusual). We simply set the number of local steps before the next communication step to be a geometric random variable of the given mean. We have in fact experimented with both this variant, and the (heuristic) version which communicates at a fixed step count, and observed very similar results. We will include results for the latter as well, as we agree that this is probably a more reasonable usage scenario in practice.

---

### Official Review · Reviewer_PXLA · 2021-07-16

**Rating:** 8
**Confidence:** 3

**Summary:**

In this paper, authors propose the asynchronous algorithm for convex optimization in distributed decentralized model. This algorithm consists of two major components - local model estimation and quantization mechanism.

**Limitations And Societal Impact:**

yes

**Main Review:**

In this paper, authors propose the asynchronous algorithm for convex optimization in distributed decentralized model that's main idea is to decrease the communication complexity of the algorithm and figure out the communication bottleneck. The idea of the algorithm is the following: we randomly select some node $i$ in the network graph G and then randomly select one of its neighbors $j$.  After this, for node $J$ we save the average of quantized values for $i$ and $j$; however, for node $i$ we also update the model estimator via the local model update. This combination of local updates together with quantization technique allows to gain in communication complexity in comparison with local SGD that was shown in the experimental part of the paper.


The presented algorithm is not restricted to some specific quantization techniques since the requirement to the Encoding and Decoding procedure is quite soft. However, it makes harder to follow the intuition without any example of applicable quantizations (only reference to the another article is provided).

Due to the Corollary 4.2 the theoretical result is close to the optimal one that is a strong point of the paper. Also, the experimental results show that the SwarmSGD algorithm works well in practice too; however, I found figures too  small and hard to read.


**Time Spent Reviewing:**

1-3 hours

---

> ### Author Response · Authors · 2021-08-10
> **Response to Reviewer 2 (PXLA)**
>
> Thank you for your review; we will follow your suggestions for improving the presentation in the next revision.

---

### Official Review · Reviewer_9ppy · 2021-07-23

**Rating:** 7
**Confidence:** 4

**Summary:**

The paper introduces SwarmSGD, a novel method that makes use of a graph of computational workers without central synchronisation. To minimize synchronisation overhead but maintain consistency between the workers' versions of the model, asynchronous, non-blocking random peer-to-peer averaging takes place every few local gradient steps, and quantization is supported.
The paper then provides theoretical analysis (under fairly restrictive assumptions), illustrating in which regime this extremely decentralised algorithm can be expected to converge (in average).
Finally, a thorough experimental section on significant benchmarks is run, demonstrating the actual potential of the method.

**Limitations And Societal Impact:**

Could lead to faster network training, which would reduce the energy footprint.

**Main Review:**

Generally speaking, the paper is clearly written, and is easy to follow. The authors provide valuable insights both in the description of the methods themselves, in the theoretical results and (perhaps most importantly) in their proofs. The Discussions part are very appreciated, and this type of analysis should become a staple for this type of paper.

The presented algorithm is novel and presents some very interesting characteristics, leading to potential gains in real-world ML workloads.

The experimental section contains very large scale experiments (at least for a paper in this theme), and demonstrate real potential.

There remains a few weaknesses in the paper, which I believe can be addressed with a bit more work.

The first one is some clarity issues.
- In the initial description of the algorithm, it's hard to understand whether the general model is quantized or not. (l65 does not reference the crucial dequantization step before applying the gradient steps)
- l85 'convergence for non-convex objectives' is not defined, making the statement harder to interpret.
- some assumptions are not explicit: being able to simply replace T by nT in the bound relies on the strong assumption that computing gradients on different devices takes the same time.
- how expensive the quantization/dequantization steps are compared to computing gradients is not explicited. That would be quite helpful to assess how useful quantization is.
- that the optimal speedup is O(sqrt(n)) is not explained. Typically, using n machines one expects the optimal speedup to be O(n). This is the case even when analyzing linearly convergent algorithms. If we were to instead rely on the dependency in T, the optimal speedup there would be O(exp(n)).

The second issue is that there appears to be some overclaiming in terms of the theoretical results. While the paper promises non-blocking  updates, these are later supposed to be atomic. Similarly, the assumption l254 that Xi_t and Enc(Xi_t) are 'simutaneous' seems like a pretty strong assumption. Of course it simplifies the analysis but the crucial question is: can the proof be carried out without it? If not, then it is an integral assumption, which should be made explicit.
The results also suffer from fairly restrictive assumptions. While not unheard of, the necessary bounds are strong restrictions.

Finally, perhaps not enough space is devoted to the experimental section.
- It would be nice to describe the exact models trained in more details. The WMT model is in particular rather elliptically referred to.
- the WMT results for SGD are quite surprising: why does the 32-devices run converge to a worse performance than the 16 one? It could conceivably be slower, but as it's got a bigger batch size should converge to a better end BLEU performance
- BLEU is referred to several times as 'accuracy'. It's not an accuracy, it's a score.
- for machine translation tasks using transformers, practitioners typically do not use SGD but Adam instead. Is there a clear path to SwarmAdam?

All told, this is a nice paper, whose potential is not quite fully exploited in its current form. I recommend acceptance, and would be willing tu upgrade my score depending on the authors' response.

**Time Spent Reviewing:**

2.5

---

> ### Author Response · Authors · 2021-08-10
> **Response to Reviewer 1 (9ppy)**
>
> We would like to thank the reviewer for the very precise review, and for the detailed feedback, which will be used to improve the clarity of our results.
> We address some of the points raised below:
>
> >  In the initial description of the algorithm, it's hard to understand whether the general model is quantized or not. (l65 does not reference the crucial dequantization step before applying the gradient steps)
> l85 'convergence for non-convex objectives' is not defined, making the statement harder to interpret.
>
> These descriptions from the introduction are indeed too concise, but they are clarified later in the paper. We will revise them in the next version.
>
> > some assumptions are not explicit: being able to simply replace $T$ by $nT$ in the bound relies on the strong assumption that computing gradients on different devices takes the same time.
>
> A weaker form of this assumption, by which the average computation time per node should be the same, should suffice for the running time translation. We can support this assumption with the batch computation time histogram (anonymized) for e.g. the ResNet50, which indeed suggests that the computation time is indeed extremely stable:
> https://www.dropbox.com/s/3sxy6270u1p42f8/ComputeHist.pdf
> (we clipped the X axis outside 240--280ms since there are no data points outside this range).
>
> We note that this transformation from $T$ to $nT$ is standard in decentralized asynchronous algorithms where operations can not be trivially grouped together, even though in practice they occur in parallel. An identical argument is used by e.g. AD-PSGD and https://arxiv.org/abs/1803.08841 to measure speedup (Both for AD-PSGD and for us activation times are random, and https://arxiv.org/abs/1803.08841 make no assumptions about activation times, thus in all cases operations can not be grouped).
>
> > how expensive the quantization/dequantization steps are compared to computing gradients is not explicited. That would be quite helpful to assess how useful quantization is.
>
> Recall that we use quantization with respect to the cubic lattice in our practical implementation, which essentially consists of taking mods coordinate-wise followed by standard stochastic quantization. (This just loses a log factor in communication relative to optimal lattices.) This is almost done “at line rate”--we measured the overhead of quantization at < 5% of the computation time for the ResNet experiments.
>
> > that the optimal speedup is $O(\sqrt{n})$ is not explained. Typically, using n machines one expects the optimal speedup to be $O(n)$.
>
> The reviewer is technically correct, but please note that speedup $\Theta(\sqrt{n})$ is the best achievable for data-parallel-SGD-type algorithms in this formulation, since we measure ergodic convergence in the non-convex setting. Subsequently, the same limitation applies to all other decentralized SGD-based algorithms. We will clarify this in the updated version.
> For convex functions, speedup is actually linear, since the convergence rate of sequential algorithms is linear as well. As far as we can tell, in the case of the algorithms which have convergence rate better than linear, the overhead from decentralization will cause the convergence to be reduced to linear at best.
>
> > While the paper promises non-blocking updates, these are later supposed to be atomic. Similarly, the assumption l254 that $X_t^i$ and $Enc(X_t^i)$ are 'simutaneous' seems like a pretty strong assumption. Of course it simplifies the analysis but the crucial question is: can the proof be carried out without it? If not, then it is an integral assumption, which should be made explicit.
>
> The reviewer is correct in that we have assumed that the writing of the quantized model average to the remote buffer happens atomically. This seems like a reasonable assumption in order to avoid very fine-grained modelling of the communication queues in the theoretical analysis, and is consistent with prior work, which assumes that message receipts or remote writes occur instantaneously, even though updates are large. For example, in AD-PSGD, it is implicitly assumed that once a communication partner starts “writing” its message at a step, it eventually completes sending that message, and that the communication buffer is never visible in “partially written” form.
>
> In practice, for large model sizes, this transmission is split into two steps: first, the node “reserves” the remote transmission slot, and then performs the remote write of the quantized model to this slot, which may not be instantaneous. If the remote node X or another concurrent interaction partner Y noticed the reserved transmission slot, then they would have to wait for the pending transmission to complete. We would argue that this amount of blocking is negligible since for instance it’s fairly unlikely that two nodes collide on a communication partner specifically precisely while the model is being transmitted. On the other hand, nodes can block for arbitrarily long during their computation without affecting the algorithms’ progress.
> However, we do acknowledge this comment, and will amend the claims to make this assumption very clear upfront.
> Please also see the response to Reviewer 4 for further discussion of the assumptions.
>
> > It would be nice to describe the exact models trained in more details. The WMT model is in particular rather elliptically referred to.
>
> We acknowledge this remark, and will add a full description for the WMT experiment. We did not insist on this, as we use the “standard” Transformer-XL model definition and parameters provided by Google https://github.com/tensorflow/models/tree/r1.11/official/transformer
>
> > the WMT results for SGD are quite surprising: why does the 32-devices run converge to a worse performance than the 16 one? It could conceivably be slower, but as it's got a bigger batch size should converge to a better end BLEU performance
>
> Please note that we count epochs as passes through the data performed _globally_ by all nodes. Specifically, at 32 nodes, at the end of an epoch, each local model has seen half the updates on average relative to a 16-node execution, so having a worse score at that point is not completely unexpected. As mentioned in the paper, running 32 nodes for some additional 50% of epochs recovers the baseline accuracy, same as 16 nodes.
>
> > BLEU is referred to several times as 'accuracy'. It's not an accuracy, it's a score.
>
> You are of course right, we will fix this.
>
> > for machine translation tasks using transformers, practitioners typically do not use SGD but Adam instead. Is there a clear path to SwarmAdam?
>
> We have also executed Swarm with the LazyAdamOptimizer, and obtained very similar convergence behaviour relative to the standard data-parallel Adam baseline. This suggests that the approach works for Adam as well, so we plan to investigate extending the analysis to adaptive methods in future work.

---

> > ### Comment · Reviewer_9ppy · 2021-08-25
> > **Thanks you for your answers**
> >
> > I've read the authors' answers in details. While I am satisfied with the justifications that are given for most assumptions, I will reiterate that I think these explanations should appear in the paper to paint the clearest possible picture. This should be easily done and I trust the author to update their final version to reflect these changes.
> >
> > Accordingly, I will update my grade to 7.
> >
> > Good work!

---

### Author Response · Authors · 2021-08-10
**Rebuttal Overview**

We would like to sincerely thank the reviewers for the high-quality feedback. We commit to following their suggestions carefully to improve our presentation in the next version of the paper, and address some questions and misunderstandings below.

---

### Decision · Program_Chairs · 2021-09-27

**Decision:**

Accept (Poster)

**Comment:**

This paper presents a gossip-style method to perform distributed optimization of a sum of individual functions, combining stochastic gradient updates with quantization mechanisms. In addition to a theoretical guarantee on the scheme's convergence, the paper reports on experiments training i) ResNets in pytorch on CIFAR-10/100 and ImageNet datasets, and a larger Transformer-XL model in tensor flow on the WMT17 dataset.
The reviews for this paper all recommend acceptance, being especially laudative about the scale of the experiments performed to validate the proposed approach. The reviews also point to an implicit assumption of atomic updates in the algorithm's operation, which the authors acknowledge and will make explicit in the final version.

The authors' experimental work is indeed quite significant. It is nice to have a theoretical guarantee; a discussion of the meaning in theory and in the experiments of the assumptions (especially the 'Bounded Local Function Variance' assumption) would be enlightening.